taxonomy and systematics/palaeontology/
environmental science

water deer, *Hydropotes inermis*, zooarchaeology,
Vietnam, Pleistocene

**Author for correspondence:**
C. M. Stimpson
e-mail: cms@wwfr.co.uk

# Confirmed archaeological evidence of water deer in Vietnam: relics of the Pleistocene or a shifting baseline?

C. M. Stimpson[1,2], S. O'Donnell[1], N. T. M. Huong[3], R. Holmes[4], B. Utting[5], T. Kahlert[1] and R. J. Rabett[1]

[1]School of Natural and Built Environment, Queen's University Belfast, Elmwood Avenue, Belfast BT7 1NN, UK
[2]Oxford University Museum of Natural History, Parks Road, Oxford OX1 3PW, UK
[3]Vietnam Academy of Social Sciences, Institute of Archaeology, 61 Phan Chu Trinh Street, Hoan Kiem, Hanoi, Vietnam
[4]School of Geography, Geology and the Environment, University of Leicester, University Road, Leicester LE1 7RH, UK
[5]Department of Archaeology, University of Cambridge, Downing Street, Cambridge CB2 3DZ, UK

CMS, 0000-0003-4327-4987; SO, 0000-0003-0731-7425;
RH, 0000-0002-6045-8705

Studies of archaeological and palaeontological bone assemblages increasingly show that the historical distributions of many mammal species are unrepresentative of their longer-term geographical ranges in the Quaternary. Consequently, the geographical and ecological scope of potential conservation efforts may be inappropriately narrow. Here, we consider a case-in-point, the water deer *Hydropotes inermis*, which has historical native distributions in eastern China and the Korean peninsula. We present morphological and metric criteria for the taxonomic diagnosis of mandibles and maxillary canine fragments from Hang Thung Binh 1 cave in Tràng An World Heritage Site, which confirm the prehistoric presence of water deer in Vietnam. Dated to between 13 000 and 16 000 years before the present, the specimens are further evidence of a wider Quaternary distribution for these Vulnerable cervids, are valuable additions to a sparse Pleistocene fossil record and confirm water deer as a component of the Upper Pleistocene fauna of northern Vietnam. Palaeoenvironmental proxies suggest that the Tràng An water deer occupied cooler, but not necessarily drier, conditions than today. We consider if the specimens represent extirpated Pleistocene populations or indicate a previously unrecognized, longer-standing southerly distribution with possible implications for the conservation of the species in the future.

# 1. Introduction

The last 2.58 Myr, the Quaternary period, saw the rise of our own species and the evolution of a modern mammal fauna [1]. Now, as the mechanisms, scale and rapidity of recent human impact on ecosystems worldwide have no geological analogue, efforts to characterize the 'Anthropocene' are gaining impetus [2,3] and the global biodiversity crisis becomes increasingly acute [4–6]. With mammal populations in decline, the necessity and utility of longer-term Quaternary records to provide 'pre-Anthropocene' insights into populations, diversity and geographical range is being demonstrated [6–11]. The principal warrant of this longer-term perspective is that baselines for quantification and description (e.g. species inventories, geographical range, habitat requirements, etc.) are set in the context of recent, ecological timescales (typically less than 100 years), where populations may have already experienced degradation through human activities (e.g. direct exploitation, habitat modification) for millennia [6,11,12].

A dilemma of this 'shifting baseline syndrome' [13,14] is that if by the time records begin, species distributions were already unrepresentative of much of the Quaternary, how effective are conservation measures likely to be in the context of degraded habitats and range [15]? In this paper, we consider a case-in-point, the water deer, *Hydropotes inermis* (Swinhoe, 1870). We present new archaeological evidence that confirms the presence of this small cervid, which has experienced pronounced recent declines in geographical range and numbers [16–18], in prehistoric Vietnam.

Water deer are currently found in eastern China and the west of the Korean peninsula (figure 1). A comprehensive review of their biology, life and natural history can be found in Schilling & Rössner [18]. In brief, this species is a small bodied, chestnut-coloured, solitary deer (Cervidae), which stands approximately 0.5 m tall at the shoulder and weighs up to 15 kg. Uniquely in the Cervidae, the males do not develop antlers but instead are characterized by long maxillary canine teeth: the current balance of evidence suggests that antler loss was secondary [18]. Water deer are almost entirely solitary beyond the rut, unlike most deer species, and thus are more akin to musk deer (Moschidae) and chevrotains (Tragulidae). Though understudied, the water deer is a potentially important model organism to investigate the evolution of the Cervidae generally and antlers, specifically [17,18,38].

Two subspecies have been recognized: the Chinese water deer *H. inermis inermis* (Swinhoe, 1870) and the Korean water deer *H. inermis argyropus* (Heude, 1884), although their validity has been questioned on molecular and morphological grounds [38–40]. Accurate estimates of current numbers of Asian water deer are difficult to determine, although the species is in decline and is classified as Vulnerable by the IUCN [16]. Chinese populations have recently experienced steep declines in numbers and a reduction in geographical range and are now present in isolated fragments in the eastern Yangtze Basin and the Zhoushan Islands (figure 1). Hunting pressure and loss of habitat are reported to be the principal causes of this decline [41].

In North Korea, water deer are presently distributed along the west coast (figure 1) and are believed to be present in most of South Korea where, in some areas, the animals are sufficiently numerous to conflict with agricultural interests [42,43]. Water deer have been the subject of reintroduction and release efforts in China and the Koreas in the twentieth and twenty-first centuries [39,43,44]. Translocated populations in the UK and France (source populations first introduced in 1900 and 1970, respectively) now account for approximately 40% of the global population [17]. Within the UK, their distribution is limited to the Midlands and East Anglia and restricted by suitable habitat availability [45]. A population was also intentionally translocated to the northeast of the Korean Peninsula by the North Korean government in the 1960s [39] and has since potentially expanded their range into the Khasanskiy district of Russia. There is currently not enough evidence to determine whether the Russian populations are transient, or stable and reproducing [46].

Historical records indicate that water deer ranged over eastern China and were found between Liaoning in the north and Guangdong in the south, as far as the lower Yangtze basin, as well as central and southern Korean peninsula [20,43] (figure 1). Bones and teeth of water deer recovered from the Holocene age (11.65 thousand years before present [BP] to recent [47], where 'present' is held at 1950) archaeological sites in the region [11,21–25] indicate an even wider former geographical distribution; as far west as eastern Tibet (Xizang), as far as Inner Mongolia in the north, to the east of the Korean Peninsula and into southern China (figure 1). Notably, recent archaeological investigations of four Iron Age sites (1300–400 years BP) have also produced evidence of water deer in central Taiwan [21]. While not all archaeological sites represent contemporaneous records, these data indicate that the water deer now occupies less than 10% of its former maximum Holocene range [11].

Earlier, Quaternary records of water deer from the Pleistocene (2.58 million to 11.65 thousand years before present [48]) are scarce and the accuracy of some fossil identifications have been queried

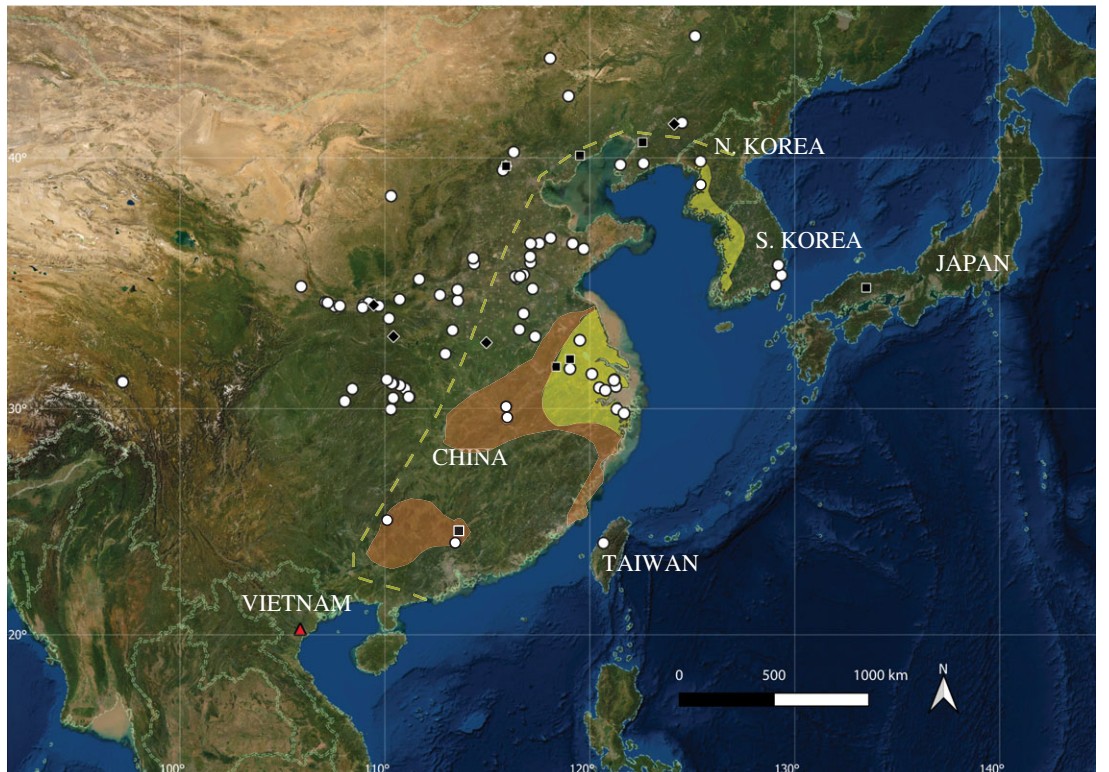

**Figure 1.** East Asia showing the location of Tràng An (red triangle: this study) and distributions of *H. inermis*: extant range (yellow shading [19]); range in twentieth century and estimated maximum westward extent of historical range (orange shading and dashed yellow line, respectively [20]); Holocene records (white circles [11,21–25]); Upper Pleistocene records (black diamonds [26–31]) and Middle Pleistocene records (black squares [26,32–36]). Data in electronic supplementary material, table S1. Map (ESRI Satellite base map, EPSG: 3857-WGS 84 Pseudo-Mercator projection) produced in QGIS [37].

(cf. [49,50]). Six Middle Pleistocene (770 000–126 000 years BP) fossil sites have yielded records across southern, central and northern China. Remarkably, records also include finds from Tarumi NT cave in western Japan where skeletal remains referred to '*Hydropotes* cf. *inermis*' were recovered with a fauna that contained extinct forms and taxa previously unrecorded from Japan [32]. Upper Pleistocene (126 000–11 650 years BP) records are limited to four sites in central China and one site in the north, near the Korean Peninsula (figure 1).

There are no recent, historical or confirmed archaeological records of water deer in Vietnam. The possibility of the prehistoric presence of the species was raised in a preliminary list of identified Upper Pleistocene animal bone in 2009 [51]. The remains were recovered by a joint Vietnamese-Bulgarian project during investigations of Mai Da Dieu (a rock shelter approx. 30 km to the south of Tràng An, the focus of the present study) in the late twentieth century. The list included 'Chinese water deer, *Hydropotes inermis* (Swinhoe, 1870)'. No figures, stratigraphic provenance, or data, however, were provided and the author points out that 'these taxonomic attributions are provisional, pending additional study' [51, p. 207]. More recently, a collection of palaeontological fossil teeth in 2020, also in northern Vietnam, from the Upper Pleistocene cave site of Lang Trang (80 000–100 000 years BP) included a single lower fourth premolar attributed to *H. inermis* [52].

Here, we present confirmed archaeological evidence of the prehistoric presence of water deer from the Tràng An Landscape Complex World Heritage Site (hereafter 'Tràng An'). The dental remains of a minimum of two individuals were recovered from late Upper Pleistocene archaeological deposits in the cave site of Hang Thung Binh 1. We present the stratigraphic and chronological context of the new specimens and the morphological and metric criteria used for taxonomic diagnosis. We then synthesize the available evidence for the environmental context of the Upper Pleistocene water deer of Tràng An. We conclude considering if the new finds reflect different ecological adaptations under different climatic conditions and are relics of the Pleistocene, or if these records are an early indication of an unrecognized southerly distribution and hence a shifted baseline, with possible implications for the conservation of the species in the future.

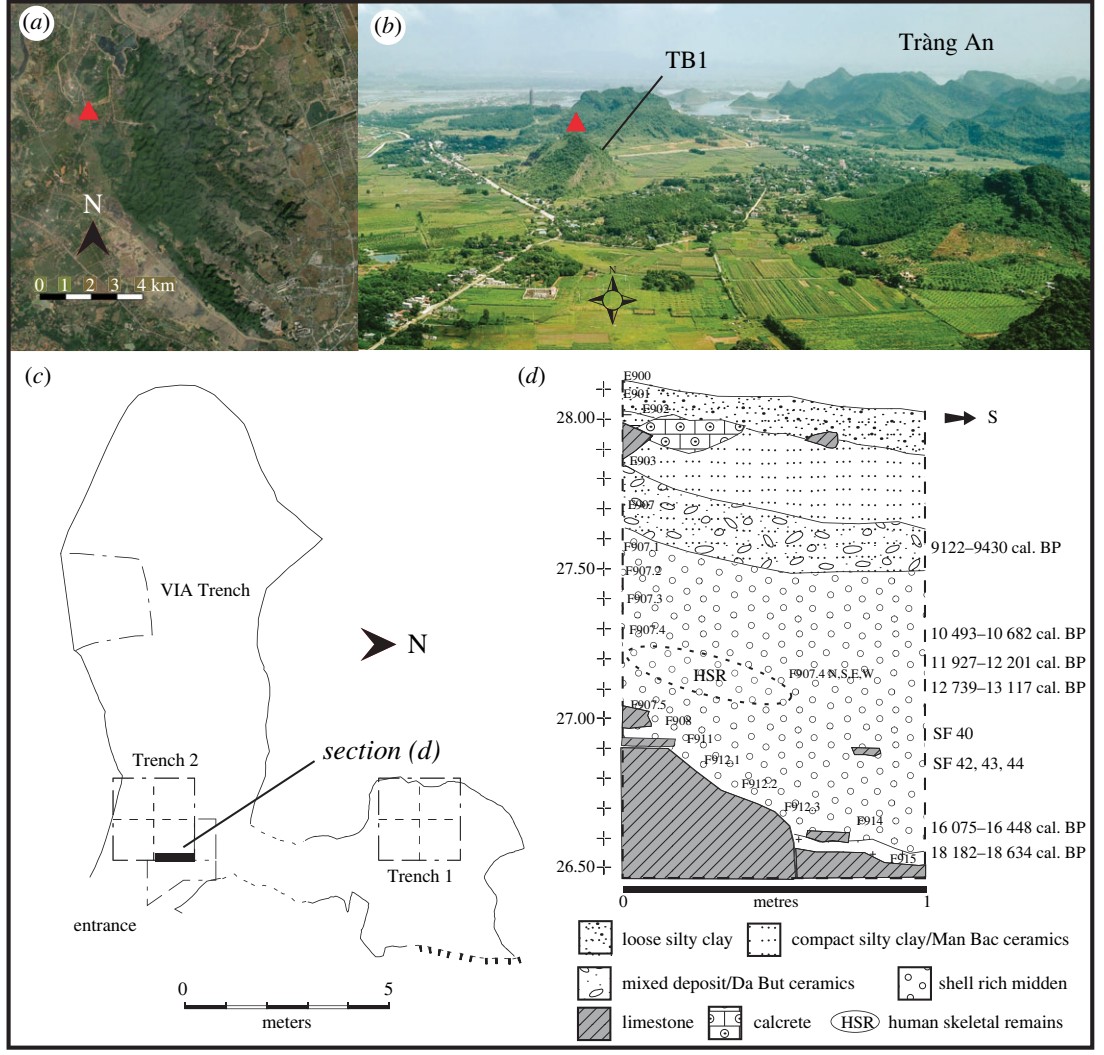

**Figure 2.** Hang Thung Binh 1. (*a*) Tràng An karst in plan, with location of Hang Thung Binh 1 (TB1—red triangle), (*b*) looking north across the coastal plain, towards isolated hill containing TB1 (red triangle) with the Tràng An karst to the east and pagoda complex in the background (original photograph: T.K.). (*c*) TB1 in plan showing the location of trenches and section D (*d*) representative west-facing section, showing calibrated radiocarbon dates (cal. BP) and find levels of *H. inermis* specimens SF40, SF42, SF43 and SF44. Levels are metres above sea level.

## 1.1. Location

The Tràng An karst landscape and environs are located in Ninh Binh Province, Northern Vietnam (see [9,53–56] for detailed descriptions). Hang Thung Binh 1 (TB1) is one of six caves within an isolated limestone hill of the same name, in the northwest corner of the Tràng An core zone, 1.5 km west of the main massif set within cultivated alluvial plains (figure 2). TB1 is a small two-chambered, east-facing cave (20°15′41.8′ N; 105°51′53.1′ E; figure 2) situated 27 m.a.s.l. and currently overlooks cultivated fields towards the northwest margin of the main massif. TB1 was investigated archaeologically in 2012 by the Vietnamese Institute of Archaeology (VIA), Hanoi [57] and by the SUNDASIA project from 2017 [56]. The SUNDASIA project excavated two trenches and the specimens reported here (table 1 and figure 3) were recovered during excavations in the second trench, which was opened in the front of the main chamber (figure 2).

## 1.2. Stratigraphic and chronological context

In Trench 2, an area of approximately 5.5 m² was excavated to an average depth of 1.60 m below the current cave floor. The trench yielded a sequence covering the twentieth century to over 18 000 years BP (figure 2). The cervid specimens were recovered from an aceramic, shell-rich (predominantly

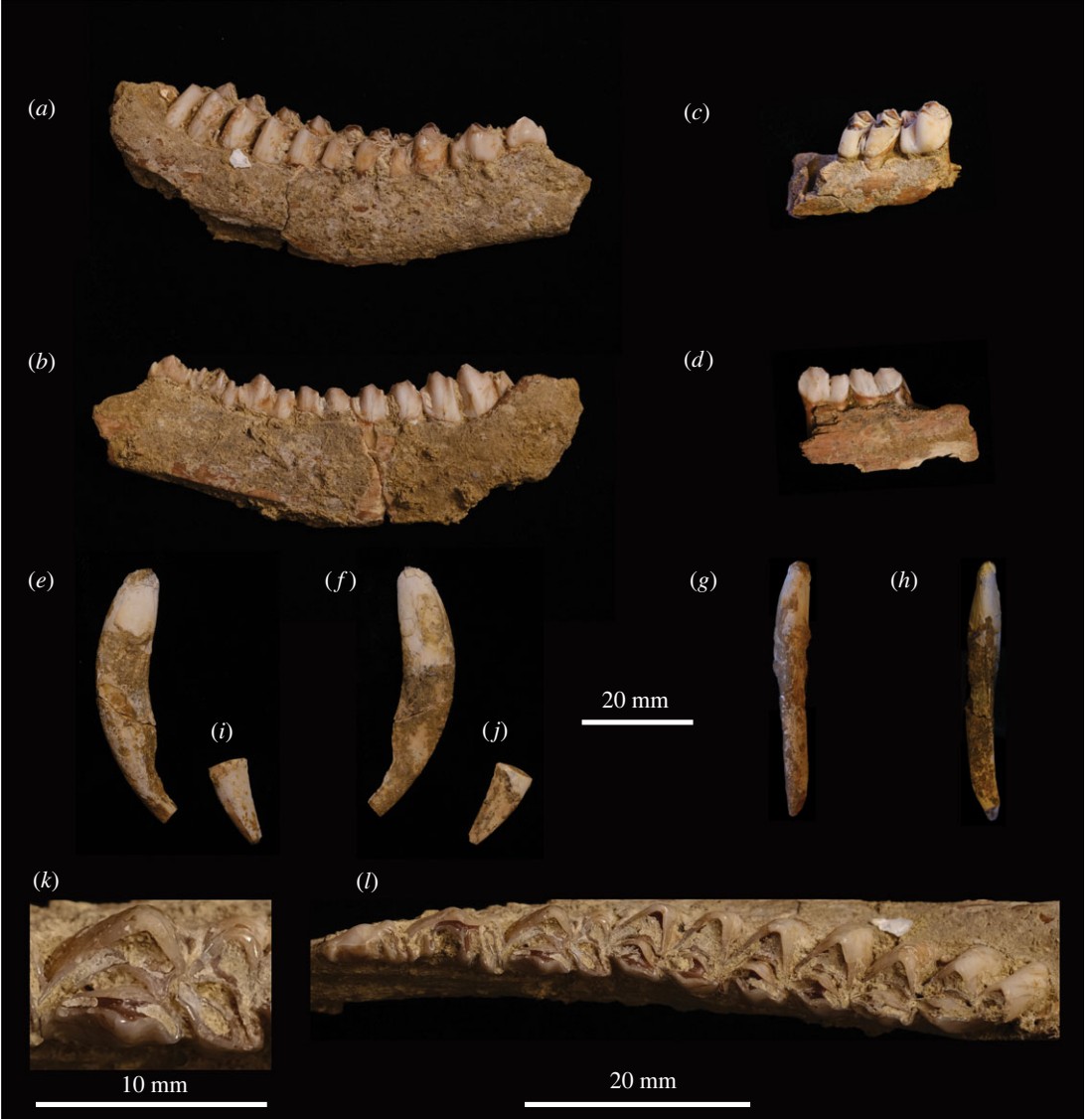

**Figure 3.** Upper Pleistocene specimens of *H. inermis* recovered from the Hang Thung Binh 1 archaeological cave site in the Tràng An World Heritage Area, Ninh Binh, Northern Vietnam. SF40 right mandible with toothrow, p2–m3, (*a*) labial side and (*b*) lingual side. SF42 right mandibular fragment with p4 and m1 *in situ*, (*c*) labial side and (*d*) lingual side. SF43 left maxillary canine, (*e*) lateral, (*f*) medial, (*g*) anterior and (*h*) posterior aspects. SF44 left maxillary canine tip, (*i*) lateral and (*j*) medial aspects. SF42 p4, (*k*) occlusal surface. SF40 toothrow, p2–m3, (*l*) occlusal surface.

**Table 1.** Provenance and description of identified specimens of *H. inermis* from the Hang Thung Binh 1 archaeological cave site in the Tràng An World Heritage Landscape Complex Site, Ninh Binh, Northern Vietnam.

| site | trench | grid sq. | context | SF no. | description |
|------|--------|----------|---------|--------|-------------|
| TB1 | 2 | TR2EE | F908.2 | 40 | right mandibular body and complete toothrow p2–m3 |
| TB1 | 2 | MS-E | F912 | 42 | right mandible fragment with p4 and m1 |
| TB1 | 2 | MS-E | F912 | 43 | left maxillary canine with root and crown |
| TB1 | 2 | MS-E | F912 | 44 | fragment of left maxillary canine -tip of crown |

*Cyclophorous* spp.), midden deposit, contexts (F907) to (F9015), which are composed of occasional stone tools, numerous fragments of animal bone and frequent charcoal fragments. Context (F907) was also notable for the presence of an inhumation [56] dating to around the Pleistocene–Holocene transition and recovered between 27.27 and 27.03 m.a.s.l. (figure 2).

Two cervid mandible fragments (TB1–F908.2–SF40 and TB1–F912–SF42, hereafter 'SF40' and 'SF42', respectively) and two fragments of maxillary canines (TB1–F912–SF43 and TB1–F912–SF44, hereafter 'SF43' and 'SF44', respectively) were recovered from the midden layers underlying the inhumation. SF40 was recovered at 26.95 m.a.s.l. on 17 November 2018. SF42, SF43 and SF44 were recovered together 6 days later in close proximity in the same area, 0.1 m deeper in the sequence at 26.85 m.a.s.l. (figure 2). There was no indication of significant reworking or discernible signs of bioturbation or disturbance to the deposits that would suggest that the specimens were intrusive or *ex situ*. From context, the specimens most likely represent the remains of animals exploited as food.

Direct dating was not attempted as previous efforts to radiocarbon date archaeological bone from TB1 and other project sites had consistently failed due to the lack of collagen content [56]. Chronological context is provided by six radiocarbon dates from charcoal recovered from the stratigraphic sequence (figure 2; electronic supplementary material, table S2). These dates were obtained via accelerator mass spectrometry at the AMS 14Chrono Centre facility at Queen's University Belfast. They were calibrated with calib 8.2 [58] using the Intcal. 20 calibration curve [59]. Calibrated radiocarbon dates are shown here as two sigma ranges as 'cal. BP' (calibrated years before present). Calibrated dates from the sequence are in superposition and indicate that the midden deposits accumulated between the end of the Last Glacial Maximum (26.5 to 19 cal. BP; [60]) and the early Holocene. The available dates effectively bracket the specimens reported here to between 13 000 and 16 000 cal. BP.

# 2. Material and methods

The identification of the specimens (table 1) was made as part of ongoing analysis of the vertebrate remains recovered from Tràng An by the SUNDASIA project. The analysis was carried out in the UK at the Oxford University Museum of Natural History with the permission of the Tràng An Management Board and Ninh Binh Peoples Committee. The specimens are to be stored and curated by the Tràng An Management Board in Ninh Binh upon completion of the project.

The specimens were identified by morphological and metric comparisons to museum specimens and morphometric data. Modern comparative specimens were consulted at the Oxford University Museum of Natural History (OUMNH) and previous observations and measurements at the Natural History Museum, UK (C.M.S., unpublished data, 2018) and American Museum of Natural History (B.U., unpublished data, 2018) were employed. Comparative descriptions and dental metric data were also compiled from the literature ([9,21,38,50,61–70]; electronic supplementary material, tables S3–S5). Dental terminology follows Bärmann & Rössner [71]. Dental measurements are shown and defined in table 2. All measurements collected in this study were taken with dial callipers to the nearest 0.01 mm.

Statistical tests were also performed to compare dental metrics of the archaeological specimens with museum specimens and published sources. Univariate normative comparisons were performed with a modified *t*-test [72–74] with step-down correction to control familywise false-positive error rate in multiple comparisons [75]. The tests were performed using the E-clip Multivariate and Univariate Normative Comparisons online platform [76]. Tests were run with a chosen $\alpha$ of 0.05. This approach assumes normality in the normative (comparative) samples. Comparative datasets were examined for departures from normality (Shapiro–Wilk $W$) using the functions in PAST v. 3.20 [77]. Output of statistical tests is shown in table 3.

# 3. Results

## 3.1. Description

SF40 is a fragment of a right mandibular body (maximum length of specimen = 78.88 mm) with a complete toothrow, p2 to m3 (figure 3). The specimen was recovered in two pieces and was conserved and then refitted—the break is located anterior of the $m^2$. Approximately 80% of the mandibular body is preserved, although the ventral surface has suffered some damage. Most of the diastema is absent, as are the incisors and pars incisiva and ascending ramus. Sediment (and a small fragment of shell on the labial side of the body, between the alveoli of the m2 and m3) has formed a hard crust through calcium carbonate deposition and is adhered to much of the specimen, particularly around the base of the teeth. After limited cleaning by hand, the morphology of the upper crowns and occlusal surfaces was clear.

**Table 2.** Measurements of three *Hydropotes inermis* specimens from the archaeological cave site of Hang Thung Binh (TB1) in the Tràng An World Heritage Area, Ninh Binh, Northern Vietnam. All measurements are in mm. Measurements from incomplete specimens are shown in brackets. C, maxillary canine; p, mandibular premolar; m, mandibular molar; *L*, length (anterior–posterior); *W*, width (labial–lingual); p2–p4, mandibular premolar row length; m1–m3, mandibular molar row length; p2–m3, mandibular toothrow length.

| tooth/toothrow | dimension | SF40 | SF42 | SF43 |
|---|---|---|---|---|
| C | *W* | / | / | 10.98 |
|  | *L* | / | / | (51.20) |
| p2 | *L* | 6.56 | / | / |
|  | *W* | 3.19 | / | / |
| p3 | *L* | 8.2 | / | / |
|  | *W* | 5.36 | / | / |
| p4 | *L* | 8.64 | 9.44 | / |
|  | *W* | 5.86 | 6.24 | / |
| m1 | *L* | 9.82 | 9.8 | / |
|  | *W* | 7.2 | 7.38 | / |
| m2 | *L* | 11 | / | / |
|  | *W* | 7.11 | / | / |
| m3 | *L* | 13.8 | / | / |
|  | *W* | 6.68 | / | / |
| p2–p4 | *L* | 24.2 | / | / |
| m1–m3 | *L* | 34.45 | / | / |
| p2–m3 | *L* | 57.96 | / | / |

The dentition is adult. Individual dental age stage (IDAS) is early IDAS 3 based on eruption and wear to the molars [78]. The p2 and p3 are relatively elongate and triangular in outline. The p4 is molarized in that, in terms of gross morphology, this tooth comprises two distinct lobes like the corresponding molars. The anterior lobe is relatively broad and consists of the mesolingual conid and antero- and posterolingual cristids detached from the transverse crest, shifted to the anterior and positioned to the lingual side of the anterior valley, which recalls an anterior fossa. A deep valley incises the labial side of the tooth at the location of the posterolabial cristid, which demarcates the anterior and posterior lobes. The posterior lobe is compressed in the anterior–posterior direction. A well-developed posterior stylid extends to the lingual side from the posterolabial conid.

The anterior and posterior lobes of the molars are broadly equal in size and shape, with triangular cusps. The adjoining edge of the posterior lobe is offset, labial of the metastylid; the preentocristid abuts the internal postprotocristid of the anterior lobe, next to the posterior margin of the anterior fossa. Except for a pointed mesostylid on the m2, which extends lingually and curves posteriorly, the lingual stylids are not well developed. Rounded mesostylids and entostylids are more prominent on the m1 and m3. The molars lack external postmetacristids. On the labial side, weakly developed anterior cingulids are present on the m1 and m2. A small, slender anterior ectostylid is present on the m3, which has a rounded hypoconulid and an isolated back fossa (figure 4).

SF42 is a small fragment (maximum length of specimen = 32 mm) of the dorsal surface of the body of a right mandible (figure 3). The ventral side is broken away. Two complete teeth, p4 and m1, are preserved *in situ* and appear slightly larger and more robust than those in SF40. A portion of the alveolus for the m2 is also preserved. As in SF40, the p4 is molarized, with an elongate posterior lobe and well-developed stylid on the posterolabial conid. A weakly developed anterior cingulid and a weakly developed and compressed (in the anterior–posterior direction) ectostylid is present on the m1. As for SF40, the joining edge of the posterior lobe is offset, labially. The lingual stylids are weakly developed and there is no external postmetacristid (figure 4).

SF43 is a left maxillary canine (maximum length of specimen = 51.20 mm; figure 3). The specimen was recovered in two pieces and refitted. The closed root and majority of crown are preserved,

**Table 3.** Summary of statistical output for tests of normality (Shapiro–Wilk $W$) and univariate normative comparisons (modified $t$-test) of specimens from Tràng An and Lang Trang with comparative data (electronic supplementary material, tables S3–S5). Significant results are shown in bold.

| spec. | measure. | taxon | n | W | p-value (norm.) | hyp | sig. | diff. | mod. t | p-value |
|---|---|---|---|---|---|---|---|---|---|---|
| SF40 | L p2–m3 | H. inermis[a] | 28 | 0.9398 | 0.1093 | 2-tailed | N | 1.778 | 1.747 | 0.084 |
| SF40 | L p2–m3 | E. cephalophus[a] | 17 | 0.9215 | 0.1566 | 1-tailed (smaller) | **Y** | **−1.972** | **−1.916** | **0.034** |
| SF40 | L p2–m3 | M. moschiferus[b] | 10 | 0.9287 | 0.4353 | 1-tailed (larger) | **Y** | **5.025** | **4.791** | **<0.001** |
| SF40 | L p2–p4 | H. inermis[a] | 24 | 0.9297 | 0.09588 | 2-tailed | N | 0.78 | 0.765 | 0.454 |
| SF40 | L p4 | E. cephalophus[a] | 7 | 0.9312 | 0.5611 | 1-tailed (smaller) | **Y** | **−3.539** | **−3.31** | **0.008** |
| SF40 | L p4 | H. inermis[a] | 7 | 0.9457 | 0.6902 | 2-tailed | N | 0.261 | 0.244 | 0.826 |
| SF42 | L p4 | H. inermis[a] | 7 | 0.9457 | 0.6902 | 2-tailed | N | 1.384 | 1.295 | 0.23 |
| SF42 | L p4 | E. cephalophus[a] | 7 | 0.9312 | 0.5611 | 1-tailed (smaller) | N | −0.545 | −0.51 | 0.294 |
| SF42 | L p4 | M. moschiferus[b] | 7 | 0.9261 | 0.518 | 1-tailed (larger) | **Y** | **6.326** | **5.917** | **<0.001** |
| SF43 | Cw | H. inermis[a] | 8 | 0.9184 | 0.4174 | 2-tailed | N | −0.061 | −0.058 | 0.943 |
| SF40 | L p2–m3 | H. inermis[a] | 28 | 0.9398 | 0.1093 | 1-tailed (larger) | **Y** | **1.778** | **1.747** | **0.047** |
| SF40 | L p2–p4 | H. inermis[a] | 24 | 0.9297 | 0.09588 | 1-tailed | N | 0.78 | 0.765 | 0.224 |
| SF42 | L p4 | H. inermis[a] | 7 | 0.9457 | 0.6902 | 1-tailed | N | 1.384 | 1.295 | 0.118 |
| SF43 | Cw | H. inermis[a] | 8 | 0.9184 | 0.4174 | 2-tailed | N | −0.061 | −0.058 | >0.9 |
| SF40 | L m3 | E. cephalophus[c] | 24 | 0.9499 | 0.2696 | 1-tailed | **Y** | **−4.038** | **−3.957** | **<0.001** |
| PIN 5792/20 | L p4 | H. inermis[a] | 7 | 0.9457 | 0.6902 | 2-tailed | **Y** | **4.276** | **4** | **<0.001** |
| PIN 5792/20 | L p4 | H. inermis[b] | 9 | 0.9318 | 0.4983 | 2-tailed | **Y** | **4.177** | **3.963** | **0.01** |

Taxon datasets: [a]Holocene.
[b]Pleistocene and Holocene.
[c]Upper Pleistocene.

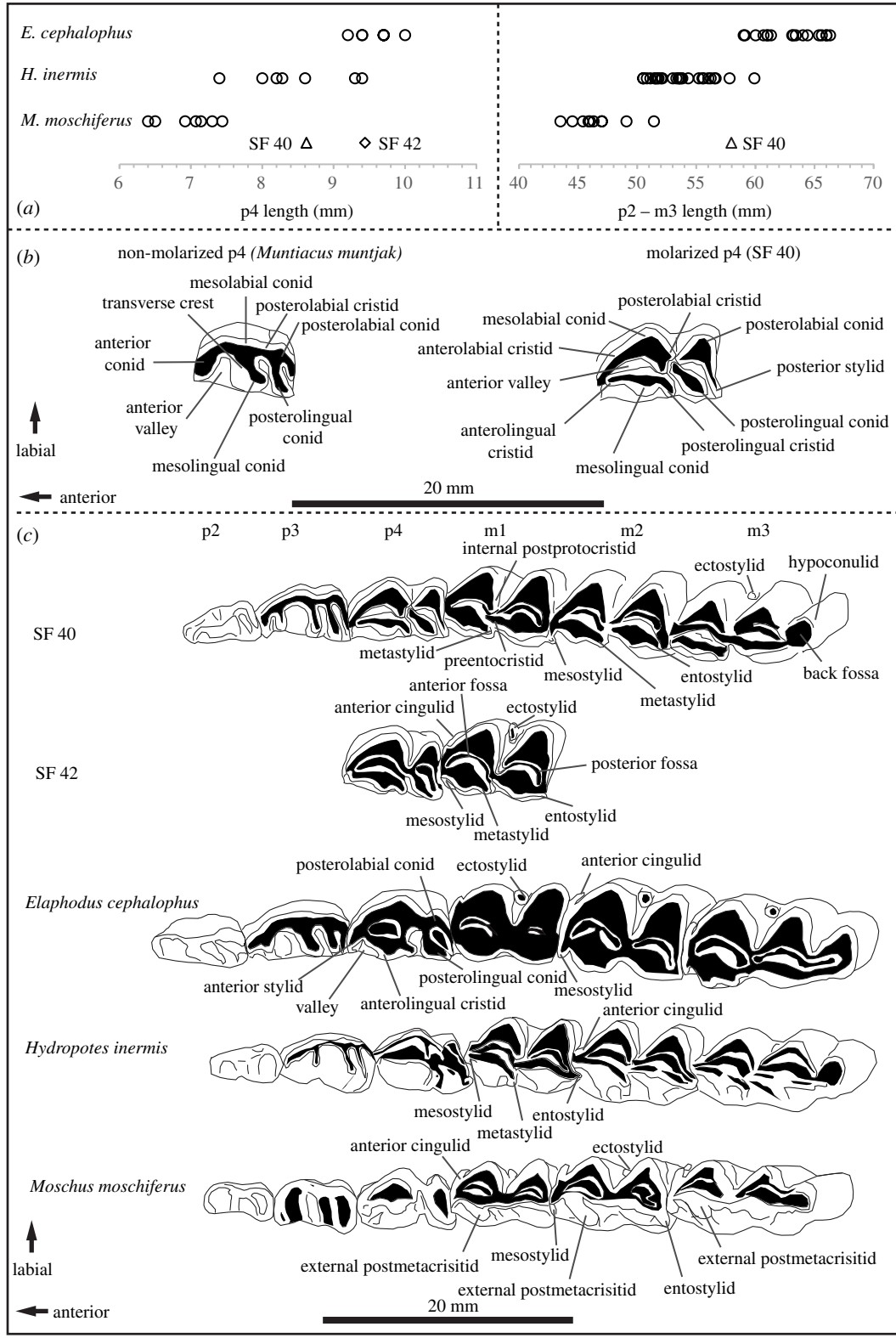

**Figure 4.** (*a*) Toothrow lengths (p2–m3) and p4 lengths of SF40 and SF42 shown with equivalent data from three taxa, *H. inermis*, *Moschus moschiferus* and *Elaphodus cephalophus* (electronic supplementary material, tables S3–S5). (*b*) Details of the occlusal surface of non-molarized and molarized fourth mandibular premolars showing characters referred to in the text. (*c*) Details of the occlusal surface of teeth of SF 40 and SF42 and the teeth of the lower jaw of comparative taxa showing characters referred to in the text. All specimens are at the individual dental age stage—IDAS 3 [78]. Specimen details: *Muntiacus muntjak* (OUMNH.ZC-20196), *Hydropotes inermis* (SF40, this study), *Moschus moschiferus* (OUMNH.ZC-2894), *H. inermis* (redrawn from [79]) and *Elaphodus cephalophus* (BMNH 92.7.13.1).

although the crown is chipped and broken at the tip. From the anterior aspect, the crown flares slightly to the lateral side and is relatively straight with a slight, sinuous line. The tooth curves posteriorly in lateral view (figure 4). The lateral surface of the crown is convex. The medial surface is relatively flat.

SF44 is the broken tip of a left maxillary canine (maximum length of the specimen = 17.92 mm; figure 3). As with SF43, SF44 is convex on the lateral side and flattened on the medial.

## 3.2. Diagnosis

The absolute size of specimens, tooth dimensions and morphology all indicate a small mesodont ruminant: either a species of the Cervidae or Moschidae.

The fourth lower premolars are present in both mandible fragments and are molarized. While Hooijer [50] describes one exceptional specimen of *M. muntjak vaginalis* (AMNH 43056), where the anterolingual cristid of the mesolingual conid adjoins the anterior conid to encircle the anterior valley (but are not fused), this character discounts *Muntiacus* [38,50,80] and indicates three candidate taxa: *Moschus* spp., *H. inermis* and *Elaphodus cephalophus* (figure 4; for *Moschus* spp., *M. moschiferus* is figured).

The smallest of the three taxa, the musk deer—*Moschus* spp.—are not true cervids and are classified in a separate family, Moschidae. *Moschus* is indicated against for SF40 by rather rounded, rectangular p2 and p3, compared to the relatively elongate and triangular equivalents in the archaeological specimen. Characteristics of the molars of *Moschus* also contrast with those of SF40 and SF42. Firstly, the lingual outlines of the molars in SF40 and SF42 are relatively simple and lack external postmetacristids. Well-developed external postmetacristids are key diagnostic characters of *Moschus* [63,80] (figure 4). Secondly, anterior cingulids and mesostylids are relatively well developed and prominent in comparative specimens from the genus, unlike the archaeological specimens. Thirdly, the location where the posterior lobe of the molars joins their anterior counterpart is much less offset and much more 'in line', lingually, with the location of the metastylid. Finally, the margins of the labial conids are rather angular, with shallow indents in the cristids, compared to the archaeological specimens, which are more rounded and lack indentations.

Univariate comparison of dental measurements from extant and Pleistocene *M. moschiferus* with the archaeological specimens found measures to be significantly larger in the archaeological specimens (table 3). Although statistical comparison was not possible due to the lack of a sufficient sample size, the width of canine SF43 was greater and the overall tooth approximately 30% larger than that of the comparative material for *Moschus*.

The largest of the three potential taxa, the tufted deer, *E. cephalophus* (17–30 kg), is currently found in southern China, with historical records from eastern Myanmar [81]. In terms of gross size and morphology, the mandibles and teeth of extant tufted deer appear larger and more robust than the TB1 material (figure 4). Unlike the archaeological material, the posterior lobes of the molars appear slightly compressed in comparison to the anterior lobes, with relatively well-developed ectostylids between them. Furthermore, the mesostylids appear well developed in comparison to metastylid and entostylid. If present, anterior cingula are very weak: they are more commonly absent in comparative material for *E. cephalophus*. The posterolabial conid of the p4 is not as developed as in the archaeological specimens. There is a relatively wide valley in the lingual wall between the anterolingual cristid of the mesolingual conid and an anterior stylid not present in SF40 and SF42 (figure 4). These characters are also consistent in large Pleistocene specimens of the genus figured in [50]. The maxillary canine in *E. cephalophus* appears shorter, but much more robust than in SF43.

Toothrow length and p4 length of SF40 was found to be significantly smaller than comparative data for extant tufted deer. No significant difference was found in p4 length between SF42 and comparative data, however, and it is likely that individual tooth length may not be the most sensitive measure to discriminate between *Hydropotes* and *Elaphodus*. Comparisons of m3 lengths of Upper Pleistocene *E. cephalophus* [70] suggest that this metric is significantly larger than SF40 (table 3).

In terms of morphology, the archaeological specimens most closely match the available comparative material for water deer, *H. inermis*. The molarized p4 in *H. inermis* has a compressed posterior lobe and a well-developed stylid on the posterolabial conid, as in both archaeological specimens (figure 4; see also [38] for the potential diagnostic significance of posterolabial conids). The metaconid of the anterior lobe is isolated from the transverse crest and shifted anteriorly to the lingual side of the anterior valley. The labial side of the tooth is deeply incised at the location of the posterolabial cristid. In the molars of examined comparative material, anterior cingulids are weak. As a rule, metastylids were the most prominent of the lingual stylids but prominent mesostylids were found to be variably present. In all cases, however, lingual stylids were not as well developed as in *Moschus*. The lingual edges of the molars are relatively simple and lack external postmetacristids, as in the archaeological specimens, a

key character to distinguish *Hydropotes* from *Moschus* [63,80]. The location of where the posterior lobe joins the anterior lobe of the molars is offset labially, as in the archaeological specimens.

While both archaeological mandibles and their teeth appear marginally larger and more robust in comparison to extant reference material for *Hydropotes*, the dental metrics of the archaeological specimens are within the ranges of equivalent available data for extant specimens (figure 4). No significant differences were found in lower tooth rows lengths or premolar row lengths between SF40 and comparative data for *H. inermis*. No differences were found between p4 lengths in SF40 and SF42 or available comparative data. No significant difference was found in canine width between SF43 and comparative data for *H. inermis* (table 3).

Confirmed Pleistocene records for *H. inermis* are scarce, as are Pleistocene age dental data (with the exception of the p4 from Lang Trang, below). It was not possible to locate a dataset to investigate the possibility of larger Pleistocene body size in the region (e.g. [50,82]). It was possible, however, to re-run the tests with the stated hypothesis that metrics from the TB1 specimens are statistically significantly larger than the available extant/Holocene data. In the case of this one-tailed test, toothrow length of SF40 was found to be significantly larger than the Holocene dataset, with $p = 0.047$; however, this result is marginal (table 3). Conversely, premolar row length in SF40 was not found to be significantly larger than the available comparative data. For SF42, the length of p4 was not found to be significantly larger the comparative dataset. Finally, canine width in SF43 was not found to be significantly larger than comparative data (table 3). The dimensions of the *H. inermis* specimen from Lang Trang (PIN 579/20; $L = 11.5$ mm; $W = 8.1$ mm [52]), however, suggest that the fourth premolar is relatively large in comparison to data from Holocene animals. Length was found to be significantly different from Holocene comparatives and when the Upper Pleistocene specimens from TB1 are included (table 3). While the p4 figured in [52] is worn, which can potentially affect measurements of teeth to a degree [82], these data suggest that the specimen from Lang Trang reflects an earlier, larger-toothed and/or larger-bodied form.

In summary, morphological and metric characters indicate that the TB1 specimens are attributable to *Hydropotes*. There are no compelling grounds to suggest a larger, Pleistocene subspecies and no reason (geographical or otherwise) for proposing a separate species. The specimens from TB1 are referred to *H. inermis*. The identified remains derive from at least two individuals. The size of the canine fragments indicates the finds include male animals (cf. [62]).

# 4. Discussion

The remains of water deer from TB1 confirm the prehistoric presence of the species in northern Vietnam. The specimens, dating to between 13 000 and 16 000 years BP, are valuable contributions to the sparse Pleistocene record for the species and are, at present, the most recent records in Vietnam.

But what do these new records from Tràng An represent? Are they simply relics of the Pleistocene or are the new finds an early indication of a previously unrealized, more recent presence in Vietnam and hence, a shifted baseline? This would not be without precedent. Work on later Quaternary palaeontological and zooarchaeological assemblages has highlighted the vulnerability and scale of range collapse of other mammalian herbivores in East and Southeast Asia (e.g. [10,11]).

In this sense, and more practically, could archaeological evidence warrant a potential case for (re)introduction into Tràng An? Reintroduction campaigns are, in practice, complex and necessarily must navigate a series of ethical, economic, sociological and ecological factors. Thorough consideration of the potential risks to the environment and livelihoods is required and must be balanced against potential benefits of ecosystem services and ecotourism, in addition to the welfare of reintroduced populations and local effects of climate change in the future. It is beyond the scope of the present paper to discuss these issues in detail. Here, we limit the discussion to a consideration of the current evidence, as a hypothetical case.

Only two other sites in Vietnam, Mai Da Dieu and Lang Trang, have yielded possible evidence of water deer. Specimens from the former locality remain unconfirmed. It is also not possible to determine any directly associated radiocarbon dates with the Mai Da Dieu material, but dates given by Popov [51] suggest a post-Last Glacial Maximum context and a broadly similar age to the TB1 specimens. The tooth reported from Lang Trang is older and currently dated to between 80 000 and 100 000 years BP [52]. In summary, both these sites are in northern Vietnam and Upper Pleistocene in age.

The Pleistocene distribution of a species is not typically perceived as a justification for designating it as 'native' within the modern area and is potentially problematic for two reasons [15]. Firstly, potential

source populations for reintroductions may derive from a different genetic population of the species, although in this case concerns about differing genetic lineages may perhaps be allayed by the questionable subspecific status of extant water deer. Secondly, given the climatic and environmental change that occurred during the Pleistocene–Holocene transition, Pleistocene populations may have been adapted to and reliant upon different conditions than the Holocene [15].

Water deer are today regarded as a temperate species (e.g. [83,84]) and extant and Quaternary records of water deer do tend towards temperate latitudes (figure 1). Observations of extant animals indicate that, while interlinked forest patches are a significant factor that can mediate localized abundance [42], water deer tend to be a lowland species with a preference for more open, marginal and riparian habitats. These habitats include reed-beds and tall, damp and undisturbed grasslands, with forbs and woody plants such as species of Asteraceae, Leguminosae and Fagaceae as favoured food plants [83,85,86]. Were similar habitats present on the plain outside the cave (figure 2), or were the Upper Pleistocene water deer of TB1 living in different conditions?

Global climates experienced significant instability between 13 000 and 16 000 years BP [87], which encompassed the glacial conditions of Heinrich Stadial 1 followed by the shift to the relatively brief period of warm and wet conditions of the Bolling-Allerod Interstadial. Chinese speleothem records indicate that this shift occurred in the region towards the end of the more recent radiocarbon date range for the water deer of TB1, between approximately 14 700 and 13 000 years BP [88]. While it is not possible, at present, to state with certainty if the Tràng An specimens derived from stadial or (at the latter end of the current date range) interstadial conditions, or a transitional period between the two, palaeoenvironmental inferences for the Tràng An karst and the intervening plains can be synthesized from a combination of proxies at various spatial scales.

Two archaeological cave sites have yielded relevant records for this time period: a multi-proxy record from Hang Trong in the interior of the Tràng An karst [55] and the pollen and spore record from Con Moong Cave [89], just over 20 km to the west of Tràng An. Proxy data from these sites included charcoal from Dipterocarpaceae, Leguminosae and Sapotaceae; pollen from euphorbs and mimosoid legumes; and stably more negative $\partial^{13}$C values (ca −30‰) of n-alkanes and n-alkanoic acids preserved in sedimentary organic carbon, indicative of forest plants using the $C_3$ photosynthetic pathway [55] and arboreal pollen assemblages comprising forest taxa adapted to cool, moist environments, such as Castanea, Castanopsis (Fagaceae), Betula (Betulaceae), Juglans, Engelhardia and Platycarya (Juglandaceae) [89]. These data suggest that limestone karsts and interleaving valleys on the southern margin of the Song Hong delta remained forested throughout the last deglaciation. Further, these areas likely acted as forest refugia during stadials and times of rapid climatic changes and supported upland vegetation typical of higher elevations today alongside lowland taxa of tropical lineages.

For the alluvial and coastal plains, basal units of two boreholes drilled into fluvio-deltaic deposits along the course of the Song Hong, east of Tràng An, yielded radiocarbon ages of 15 000–13 000 cal. BP [90,91] as well as palaeoenvironmental proxies in the form of palynological assemblages and sedimentological structures [92]. Facies 1.1 of the ND-1 core, located approximately 30 km ENE from Tràng An, produced two radiocarbon ages of 15 000 and 14 800 cal. BP [90]. The lithology of this unit was upward-fining and consisted of cross-bedded fluvial sands, and interlaminated muds and organic sediments interpreted as lateral accretion within a river meander, as well as channel fill. Pollen preservation was relatively poor but contained grains of conifers (Cryptomeria; Taxodioideae; Pinus), upland broadleaf taxa (e.g. Quercus) and various temperate riparian elements [92].

Further downstream, near to the modern delta front and about 70 km east from Tràng An, Unit 1 of the NP core sits underlies a radiocarbon age of around 12 000 cal. BP [91]. The silts of this unit yielded pollen assemblages comprising grasses (Poaceae), sedges (Cyperaceae), willow (Salix), oak (Quercus) and chinquapin (Castanopsis). These assemblages suggest a floodplain environment with scattered freshwater marsh, with the pollen from temperate broadleaf elements likely transported to the site fluvially from the surrounding uplands. Given the depths of these basal units, the riparian, floodplain and marsh environments they represent occurred within the ancient valley incised by the Song Hong during times of lower sea level in the Upper Pleistocene. It is a reasonable proposition then, that the plain outside TB1 carried a seaward-draining watercourse that flowed through a similar mosaic of alluvial environments, which incorporated more open wetland habitats, surrounded by upland and limestone forests on the karst.

The available proxies suggest that similar or preferred habitats of extant water deer were present in the Upper Pleistocene on the plains around TB1, which reflected cooler, but not necessarily drier, conditions than the subtropical climate of the Holocene. Regional palaeoenvironmental proxies also indicate a drier and cooler climate at the beginning of the Holocene, following the Younger Dryas stadial, which saw reduced temperature and precipitation approximately 12 900–11 700 years BP.

While water deer may have persisted into the early Holocene, increasing temperature and humidity and the establishment of subtropical climate and environmental change [93] may have precluded the persistence of populations of water deer in Northern Vietnam.

This argument, at least in part, would be an artefact of the shifting baseline syndrome. The historical distribution of water deer in China [20] and the recent finds in Taiwan [21] present a significant challenge to this simplistic climatic and environmental narrative. Historical records place water deer at subtropical latitudes in China until recently, in the twentieth century. In addition to the identified archaeological material, historical documents indicate that water deer were, until relatively recently, common and widespread in Taiwan until extirpation in the nineteenth century [21].

In practical terms, only confirmed and reliably dated specimens would provide unequivocal evidence of populations surviving into the Holocene. A search for potential specimens from Holocene age archaeological sites in Vietnam, either in archives or during future investigations would be worthwhile, as would a review of potential documentary evidence. A comprehensive assessment would also require other factors such as hunting by humans, or competitive exclusion by other species to be considered, but there are currently no clear climatic or environmental factors that would have been a definitive barrier to the survival of water deer into the Holocene in northern Vietnam.

# 5. Conclusion

Newly identified fossils from the archaeological cave site of Hang Thung Binh 1 in the Tràng An World Heritage Landscape Complex Site confirm the presence of water deer, *H. inermis*, in northern Vietnam in the Upper Pleistocene. The new specimens are further evidence of a wider Quaternary distribution for these Vulnerable, globally declining cervids, are valuable additions to a sparse Pleistocene fossil record and confirm water deer as a component of the Upper Pleistocene fauna of northern Vietnam.

Current archaeological and palaeontological evidence for the presence of water deer in the country is extremely sparse and restricted to the Upper Pleistocene. While water deer are today associated with temperate latitudes, a brief survey of the known historical distribution and recent archaeological evidence indicates that the species has been capable of surviving in subtropical climates and habitats. The possibility that water deer survived into the Holocene in northern Vietnam is a hypothesis to be tested.

Data accessibility. The data are provided in electronic supplementary material [94].

Authors' contributions. C.M.S. and R.J.R. conceived the study; C.M.S. conducted analyses and identification and drafted the manuscript with contributions from all authors; C.M.S., R.H. and B.U. conducted the review of Quaternary records of *Hydropotes inermis*; C.M.S. and B.U. performed statistical tests; T.K. translated German language source material; C.M.S., B.U., R.J.R. and T.K. carried out excavation, site analysis and recording; S.O. and N.T.M.H. produced the palaeoenvironmental synthesis. All authors revised the manuscript and gave final approval for publication.

Competing interests. We declare we have no competing interests.

Funding. Funding for this project was provided by the Arts and Humanities Research Council (Global Challenges Research Fund) award no. AH/N005902/1 to RR, a UKRI COVID-19 Grant Extension Allocation award and through the support of the Xuan Truong Construction Enterprise.

Acknowledgements. We thank the Tràng An Management Board, the Ninh Binh Provincial People's Committee and Xuan Truong Construction Enterprise for ongoing support for research in Tràng An. Dr Alex Wilshaw cleaned and conserved the archaeological material in the field. We thank all our colleagues from the SUNDASIA project field teams, Ninh Binh People's Museum, Vietnamese Institute of Archaeology (Hanoi) and Tràng An Management Board. C.M.S. would like to acknowledge the assistance and support of Eileen Westwig, Mark Carnall, Eliza Howlett and the Oxford University Museum of Natural History. The study benefited from access to the comparative museum collections, as well as those of the American Museum of Natural History and Natural History Museum, UK. We thank two anonymous reviewers for their encouraging comments and constructive criticism, which improved the content and structure of the manuscript.

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
