## [Peer Review File · Royal Society Open Science]

Review History

RSOS-210529.R0 (Original submission)

Review form: Reviewer 1

Is the manuscript scientifically sound in its present form?

Yes

Are the interpretations and conclusions justified by the results?

Yes

Is the language acceptable?

Yes

Do you have any ethical concerns with this paper?

No

Have you any concerns about statistical analyses in this paper?

No

Recommendation?

Accept with minor revision (please list in comments)

Comments to the Author(s)

Stimpson et al. have presented new specimens from Hang Thung Binh 1 cave, Vietnam, which can be attributed to the water deer, *Hydropotes inermis*. These novel late Pleistocene occurrences of this species represents the southern-most extent of the range, both extant and fossil, and this represents an important contribution to our knowledge of the biogeography of this taxon. The authors have made adequate comparisons to extant cervoids from the region and have provided a cogent argument for why these specimens represent water deer. The Quaternary fossil record of Southeast Asia remains largely understudied compared to East Africa or North America, and this study provides another piece of the puzzle of how extant mammal communities in the region have evolved.

I thought the study was very interesting, since not much is known about fossil water deer. It's great to see some basic taxonomic work from this region, couched within the larger topic of shifting baselines and Quaternary extinctions and extirpations. I thought the morphological and statistical analyses were quite sufficient to determine whether the specimens under investigation were in fact water deer. My biggest concern was with the structure of the discussion. It was a little hard to follow, and did not seem, to me at least, to flow well in an "answering research questions" framework. This is not to say that the discussion needs a re-write, but I would say that a re-structuring would benefit the reader.

For example, lines 378-380 seem to present a good question that can be addressed with your new data, and so do lines 412-414. I think leading with what you found, i.e., Vietnamese water deer in the latest Pleistocene, and how that can inform the idea of shifting baselines or Quaternary extinctions/extirpations would provide a more impactful discussion. It would also harken back to your introduction where you introduce these ideas at the beginning. Your discussion about the changing floral regimes through the late Pleistocene and Holocene could follow, providing supporting arguments for why the water deer are found in their extant range. While reading the paper, it wasn't really clear to me how the climatic changes that occurred regionally, i.e. Heinrich event 1, etc, were related to the patterns of vegetation change, and therefore the water deer. It might be worth simplifying that section and making it more streamlined.

I also think the manuscript would benefit from a discussion about Quaternary extinctions in the region, and importantly faunal persistence, range shifts, and metapopulation dynamics. What you have seems to be a relict Pleistocene population that got extirpated. There's a lot of exciting research coming out of the region such as the giant muntjac that you described from Northern Vietnam, the work Sam Turvey has done on population collapses in muntjacs, and the new species of gibbon from China. It would help show that there is still much to be learnt from Pleistocene and Holocene records from this part of the world. Moreover, it would be hugely beneficial to see what other species in the region show similar patterns. I know off-hand that all three species of asian rhinos were once more widely distributed through South Asian, Southern China, and Southeast Asia, but are now relegated to isolated pockets.

Ending with your conservation and re-introduction message would in my mind be a good way to wrap up this fascinating story. Overall, I think this is a great paper. I had additional minor edits, that I have included in the attached pdf (Appendix A).

Review form: Reviewer 2

Is the manuscript scientifically sound in its present form?

Yes

Are the interpretations and conclusions justified by the results?

Yes

Is the language acceptable?

Yes

Do you have any ethical concerns with this paper?

No

Have you any concerns about statistical analyses in this paper?

No

Recommendation?

Accept with minor revision (please list in comments)

Comments to the Author(s)

Dear authors,

I have read your manuscript, titled "Confirmed archaeological evidence of water deer in Vietnam: relics of the Pleistocene or a shifting baseline?", on evidence of the first Vietnamese *Hydropotes* population with much interest and pleasure. This is a clearly written transdisciplinary manuscript based on a carefully conducted study that not only bridges fields palaeontology/archaeology (past) and conservation biology (present and future) in a model manner, but also guides to future management of conservation measures for vulnerable species and hints to the mandatory consideration of the big picture as a necessity for reasonable procedures. On the other hand, it provides as well valuable additional information to mammal zooarchaeologists and palaeontologists on the fossil record of *Hydropotes inermis* and its ancient habitat. Your methodology is rigorous and the conclusions are fully supported by the data. I congratulate on the comparative morphology of the teeth among Pecora, which is really not an easy task, but you managed that with bravery.

In my eyes the manuscript is almost ready for publication and has major importance to science in the framework described above. The only criticism I come up with concerns the description of the tooth crown morphology. Although the p4 represents a key tooth for species identification, p2 and p3 hold cervid specific differences from *Moschus*, but aren't described, and m1 to m3 are almost not described. This would be important to future work searching for more *Hydropotes* material in archives and excavations. Please see more detailed comments in the annotated manuscript itself. Moreover, you say that you have used Bärmann & Rössner (2011) as the source for tooth crown element nomenclature, what is only true in part. Some elements are differently named than in B & R and do not even follow the principles given there. Hence, either please adapt or clarify in the text that B & R has been used as a basis for your modifications. The molarisation of p4s is a commonly known phenomenon in mammal odontology and mammal palaeontology what has not to be put in quotation marks. Please see for example Janis and Lister (1985) (see Appendix B). However, although homology of crown elements in non-molarised and molarised is not evidenced in the fossil record by step by step transition states, there are ideas / hypotheses on what element in a non-molarised p4 is homologous to an element in a molarised p4. As to cervids I have provided more details in the annotated manuscript (see Appendix C). Last but not least, as tooth morphology is so crucial in that paper, higher magnified occlusal views are recommended.

As I wrote at the beginning, I have read the ms with pleasure and look very much forward to seeing it published.

Decision letter (RSOS-210529.R0)

Dear Dr Stimpson

On behalf of the Editors, we are pleased to inform you that your Manuscript RSOS-210529 "Confirmed archaeological evidence of water deer in Vietnam: relics of the Pleistocene or a shifting baseline?" has been accepted for publication in Royal Society Open Science subject to minor revision in accordance with the referees' reports. Please find the referees' comments along with any feedback from the Editors below my signature.

Please submit your revised manuscript and required files (see below) no later than 7 days from today's (ie 26-May-2021) date. Note: the ScholarOne system will 'lock' if submission of the revision is attempted 7 or more days after the deadline. If you do not think you will be able to meet this deadline please contact the editorial office immediately.

on behalf of Dr Emily Lindsey (Associate Editor) and Pete Smith (Subject Editor)
openscience@royalsociety.org

Associate Editor Comments to Author (Dr Emily Lindsey):

Associate Editor: 1

Comments to the Author:

The article was reviewed by two experts, both of whom noted the interest and value of the article and recommended publishing with minor revisions. The two main concerns the reviewers raised that should be addressed for publication were:

1. A more well-developed description of the tooth morphology, and ensuring accuracy and consistency with the diagnostic references.
2. A restructuring of the discussion for clarity and highlighting the importance of the presented study for broader research and conservation-related questions.

Both reviewers also made substantive in-line comments in the manuscript .pdf's that the authors should address as well.

Congratulations on a well-received article! I look forward to the revised submission.

Reviewer comments to Author:

Reviewer: 1

Comments to the Author(s)

Stimpson et al. have presented new specimens from Hang Thung Binh 1 cave, Vietnam, which can be attributed to the water deer, *Hydropotes inermis*. These novel late Pleistocene occurrences of this species represents the southern-most extent of the range, both extant and fossil, and this represents an important contribution to our knowledge of the biogeography of this taxon. The authors have made adequate comparisons to extant cervoids from the region and have provided a cogent argument for why these specimens represent water deer. The Quaternary fossil record of Southeast Asia remains largely understudied compared to East Africa or North America, and this study provides another piece of the puzzle of how extant mammal communities in the region have evolved.

I thought the study was very interesting, since not much is known about fossil water deer. It's great to see some basic taxonomic work from this region, couched within the larger topic of shifting baselines and Quaternary extinctions and extirpations. I thought the morphological and statistical analyses were quite sufficient to determine whether the specimens under investigation were in fact water deer. My biggest concern was with the structure of the discussion. It was a little hard to follow, and did not seem, to me at least, to flow well in an "answering research questions" framework. This is not to say that the discussion needs a re-write, but I would say that a re-structuring would benefit the reader.

For example, lines 378-380 seem to present a good question that can be addressed with your new data, and so do lines 412-414. I think leading with what you found, i.e., Vietnamese water deer in the latest Pleistocene, and how that can inform the idea of shifting baselines or Quaternary extinctions/extirpations would provide a more impactful discussion. It would also harken back to your introduction where you introduce these ideas at the beginning. Your discussion about the changing floral regimes through the late Pleistocene and Holocene could follow, providing supporting arguments for why the water deer are found in their extant range. While reading the paper, it wasn't really clear to me how the climatic changes that occurred regionally, i.e. Heinrich event 1, etc, were related to the patterns of vegetation change, and therefore the water deer. It might be worth simplifying that section and making it more streamlined.

I also think the manuscript would benefit from a discussion about Quaternary extinctions in the region, and importantly faunal persistence, range shifts, and metapopulation dynamics. What you have seems to be a relict Pleistocene population that got extirpated. There's a lot of exciting research coming out of the region such as the giant muntjac that you described from Northern Vietnam, the work Sam Turvey has done on population collapses in muntjacs, and the new species of gibbon from China. It would help show that there is still much to be learnt from Pleistocene and Holocene records from this part of the world. Moreover, it would be hugely beneficial to see what other species in the region show similar patterns. I know off-hand that all

three species of asian rhinos were once more widely distributed through South Asian, Southern China, and Southeast Asia, but are now relegated to isolated pockets.

Ending with your conservation and re-introduction message would in my mind be a good way to wrap up this fascinating story. Overall, I think this is a great paper. I had additional minor edits, that I have included in the attached pdf.

Reviewer: 2

Comments to the Author(s)

Dear authors,

I have read your manuscript, titled "Confirmed archaeological evidence of water deer in Vietnam: relics of the Pleistocene or a shifting baseline?", on evidence of the first Vietnamese *Hydropotes* population with much interest and pleasure. This is a clearly written transdisciplinary manuscript based on a carefully conducted study that not only bridges fields palaeontology/archaeology (past) and conservation biology (present and future) in a model manner, but also guides to future management of conservation measures for vulnerable species and hints to the mandatory consideration of the big picture as a necessity for reasonable procedures. On the other hand, it provides as well valuable additional information to mammal zooarchaeologists and palaeontologists on the fossil record of *Hydropotes inermis* and its ancient habitat. Your methodology is rigorous and the conclusions are fully supported by the data. I congratulate on the comparative morphology of the teeth among Pecora, which is really not an easy task, but you managed that with bravery.

In my eyes the manuscript is almost ready for publication and has major importance to science in the framework described above. The only criticism I come up with concerns the description of the tooth crown morphology. Although the p4 represents a key tooth for species identification, p2 and p3 hold cervid specific differences from *Moschus*, but aren't described, and m1 to m3 are almost not described. This would be important to future work searching for more *Hydropotes* material in archives and excavations. Please see more detailed comments in the annotated manuscript itself. Moreover, you say that you have used Bärmann & Rössner (2011) as the source for tooth crown element nomenclature, what is only true in part. Some elements are differently named than in B & R and do not even follow the principles given there. Hence, either please adapt or clarify in the text that B & R has been used as a basis for your modifications. The molarisation of p4s is a commonly known phenomenon in mammal odontology and mammal palaeontology what has not to be put in quotation marks. Please see for example Janis and Lister (1985) (attached). However, although homology of crown elements in non-molarised and molarised is not evidenced in the fossil record by step by step transition states, there are ideas / hypotheses on what element in a non-molarised p4 is homologous to an element in a molarised p4. As to cervids I have provided more details in the annotated manuscript. Last but not least, as tooth morphology is so crucial in that paper, higher magnified occlusal views are recommended. As I wrote at the beginning, I have read the ms with pleasure and look very much forward to seeing it published.

===PREPARING YOUR MANUSCRIPT===

===PREPARING YOUR REVISION IN SCHOLARONE===

- If you are providing image files for potential cover images, please upload these at this step, and inform the editorial office you have done so. You must hold the copyright to any image provided.
- A copy of your point-by-point response to referees and Editors. This will expedite the preparation of your proof.

- Ensure that your data access statement meets the requirements at <https://royalsociety.org/journals/authors/author-guidelines/#data>. You should ensure that you cite the dataset in your reference list. If you have deposited data etc in the Dryad repository, please only include the 'For publication' link at this stage. You should remove the 'For review' link.
- If you are requesting an article processing charge waiver, you must select the relevant waiver option (if requesting a discretionary waiver, the form should have been uploaded at Step 3 'File upload' above).
- If you have uploaded ESM files, please ensure you follow the guidance at <https://royalsociety.org/journals/authors/author-guidelines/#supplementary-material> to include a suitable title and informative caption. An example of appropriate titling and captioning may be found at https://figshare.com/articles/Table_S2_from_Is_there_a_trade-off_between_peak_performance_and_performance_breadth_across_temperatures_for_aerobic_scope_in_teleost_fishes_/3843624.

Author's Response to Decision Letter for (RSOS-210529.R0)

See Appendices D & E.

Decision letter (RSOS-210529.R1)

Dear Dr Stimpson,

I am pleased to inform you that your manuscript entitled "Confirmed archaeological evidence of water deer in Vietnam: relics of the Pleistocene or a shifting baseline?" is now accepted for publication in Royal Society Open Science.

on behalf of Dr Emily Lindsey (Associate Editor) and Pete Smith (Subject Editor)
openscience@royalsociety.org

Associate Editor Comments to Author (Dr Emily Lindsey):
Associate Editor: 1
Comments to the Author:
(There are no comments.)

Reviewer comments to Author:

Appendix A**ROYAL SOCIETY
OPEN SCIENCE****Confirmed archaeological evidence of water deer in
Vietnam: relics of the Pleistocene or a shifting baseline?**

Journal:	Royal Society Open Science
Manuscript ID	RSOS-210529
Article Type:	Research
Date Submitted by the Author:	28-Mar-2021
Complete List of Authors:	Stimpson, Christopher; Queen's University Belfast School of Natural And Built Environment; Oxford University Museum of Natural History O'Donnell, Shawn; Queen's University Belfast, School of Natural and Built Environment Mai Huong, Nguyen Thi; Vietnam Academy of Social Sciences, Institute of Archaeology Holmes, Rachael; University of Leicester, School of Geography, Geology and the Environment Utting, Benjamin; University of Cambridge, Archaeology Kahlert, Thorsten; Queen's University Belfast Rabett, Ryan; Queen's University Belfast
Subject:	taxonomy and systematics < BIOLOGY, palaeontology < BIOLOGY, environmental science < BIOLOGY
Keywords:	water deer, Hydropotes inermis, zooarchaeology, Vietnam, Pleistocene
Subject Category:	Ecology, Conservation, and Global Change Biology

Author-supplied statements

Relevant information will appear here if provided.

Ethics

Does your article include research that required ethical approval or permits?:

This article does not present research with ethical considerations

Statement (if applicable):

CUST_IF_YES_ETHICS :No data available.

Data

It is a condition of publication that data, code and materials supporting your paper are made publicly available. Does your paper present new data?:

Yes

Statement (if applicable):

The datasets supporting this article have been uploaded as electronic supplementary material.

Conflict of interest

I/We declare we have no competing interests

Statement (if applicable):

CUST_STATE_CONFLICT :No data available.

Authors' contributions

This paper has multiple authors and our individual contributions were as below

Statement (if applicable):

CMS and RR conceived the study; CMS conducted analyses and identification and drafted the manuscript with contributions from all authors; CMS, RH and BU conducted the review of Quaternary records of *Hydropotes inermis*; CMS and BU performed statistical tests; TK translated German language source material; CMS, BU, RR, TK, carried out excavation, site analysis and recording; SO and NTMH produced the paleoenvironmental synthesis. All authors revised the manuscript and gave final approval for publication.

[revised manuscript text omitted]

The dentition is adult but is unworn. Individual dental age stage (IDAS) is early IDAS
3 based on eruption and wear to the molars [78]. The p4 is “molarised” in that, in terms of
gross morphology, this tooth comprises of two distinct lobes. The anterior lobe is broad,
whereas the posterior lobe is compressed and elongate in the labial-lingual direction with
well-developed posterolingual and posterolabial conids. The anterior and mesolingual conids
are fused, forming a contiguous structure, which enclose an anterior fossette. The lingual
edges of the metaconids of the molars are simple, with single shallow folds. Weakly
developed anterior cingulids are present on the m1 and m2. A small, slender anterior
ectostylid is present on the m3 (**figure 4**).

SF42 is a small fragment (maximum length of specimen = 32 mm) of the dorsal surface
of the body of a right mandible (**figure 3**). The ventral side is broken away. Two complete,
unworn teeth, p4 and m1, are preserved in situ and appear slightly larger and more robust
than those in SF40. A portion of the alveolus for the m2 is also preserved. As in SF40, the p4
is also “molarised” (**figure 4**), with an elongate posterior lobe and well-developed
posterolingual and posterolabial conids. A weakly-developed anterior cingulid and a weakly-
developed and elongate (in the labial-lingual direction) ectostylid is present on the m1. The
lingual edge of the metaconids lack complex folds or crenulations.

SF43 is a left maxillary canine (maximum length of specimen = 51.20 mm; **figure 3**).
The specimen was recovered in two pieces and refitted. The closed root and majority of crown
are preserved, although the crown is chipped and broken at the tip (**figure 4**). The tooth curves
posteriorly. The lateral surface of the crown is concave. The medial surface is relatively flat.

SF44 is the broken tip of a left maxillary canine (maximum length of specimen = 17.92
232 mm; **figure 3**). As with SF43, SF44 is convex on the lateral side and flattened on the medial.

4.2 Diagnosis

Absolute size of specimens, tooth dimensions and morphology all indicate small ruminant: either a species of the Cervidae or Moschidae. The relatively long, posteriorly curved canine with a closed root also indicates the so-called “fanged” deer or musk deer.

The fourth lower premolars are present in both mandible fragments and are “molarised”. While Hooijer [51] describes one exceptional specimen of *M. muntjak vaginalis* (AMNH 43056), where the anterior and mesolingual conids meet (but are not fused), this character discounts *Muntiacus* [19, 51, 80] and indicates three candidate taxa: *Moschus* spp., *H. inermis* and *Elaphodus cephalophus* (**figure 4**; for *Moschus* spp., *M. moschiferus* is figured).

The smallest of the three taxa, the musk deer - *Moschus* spp. - are not true cervids and are classified in a separate family, Moschidae. Musk deer are currently distributed from the Himalayas to North East Asia, with up to seven extant species [81]. The smallest species, *M. berezovskii* (Flerov, 1928; extant body weight 6-9 kg), the “dwarf” or “forest musk deer”, ranges into South China and marginally into north Vietnam, where it has been reported in karst habitats. The much-reduced Vietnamese populations were, until recently, thought to be Siberian musk deer, *M. moschiferus* [82].

Moschus is indicated against here by the relatively simple lingual outline of the molars in SF40 and SF42. Specifically, “double-folded” lingual margins of the metaconids are key diagnostic characters of *Moschus* [63,80] and are absent in the archaeological material (**figure 4**). Furthermore, anterior cingulids and mesostylids are relatively well developed in comparative specimens from the genus in comparison to the archaeological material. Conversely, posterolingual and posterolabial conids were well developed on the p4 in the available reference material.

Univariate comparison of dental measurements from extant and Pleistocene *M. moschiferus* with the archaeological specimens, found measures to be statistically significantly larger in the archaeological specimens (**table 3**). Although statistical comparison was not possible due to lack of a sufficient sample size, the width of canine SF43 was greater and the overall tooth approximately 30% larger than that of the comparative material for *Moschus*.

The largest of the three potential taxa, the tufted deer, *E. cephalophus*, (17-30kg) is currently found in southern China, with historical records from eastern Myanmar [83]. In terms of gross size and morphology, the mandibles and teeth of extant tufted deer appear larger and more robust than the TB1 material (**figure 4**). Relatively well developed ectostylids are apparent on the molars. If present, anterior cingula are very weak: they are more commonly absent in comparative material for *E. cephalophus*. The posterolabial and posterolingual conids of the p4 are not as developed as in the archaeological specimens, but there is a relatively deep fold in the lingual wall between the anterior stylid and anterior conid, which is absent in the p4 of SF40 and SF42. These characters are also consistent in large Pleistocene specimens of the genus figured in [51]. The maxillary canine in *E. cephalophus* appears shorter, but much more robust than in SF43.

Toothrow length and p4 length of SF40 was found to be statistically significantly smaller than comparative data for extant tufted deer. No statistically significant difference was found in p4 length between SF42 and comparative data, however, and it is likely that individual tooth length may not be the most sensitive measure to discriminate between *Hydropotes* and *Elaphodus*. Comparisons of m3 lengths of Upper Pleistocene *E. cephalophus* [70] suggest that this metric is statistically significantly larger than SF40 (**table 3**).

In terms of morphology, the archaeological specimens most closely match the
available comparative material for water deer, *H. inermis*. The molarised p4 in *H. inermis* has
an elongate posterior lobe and well-developed posterolingual and posterolabial conids as in
both archaeological specimens (**figure 4**; see also [19] for the potential diagnostic significance
of posterolabial conids). Weakly developed anterior cingulids are present on the molars of
available comparative material. The lingual walls of the metaconids are relatively simple (i.e.,
lack complex folds), as in the archaeological specimens, a key character to distinguish
*Hydropotes* from *Moschus* [63,80].

While both archaeological mandibles and their teeth appear marginally larger and
more robust in comparison to extant reference material for *Hydropotes*, the dental metrics of
the archaeological specimens are within the ranges of equivalent available data for extant
specimens (**figure 4**). No statistically significant differences were found in lower tooth rows
lengths or premolar row lengths between SF40 and comparative data for *H. inermis*. No
differences were found between p4 lengths in SF40 and SF42 or available comparative data.
No significant difference was found in canine width between SF43 and comparative data for
*H. inermis* (**table 3**).

Confirmed Pleistocene records for *H. inermis* are scarce, as are Pleistocene age dental
data (with the exception of the p4 from Lang Trang, below). It was not possible to locate a
data set to investigate the possibility of larger Pleistocene body size in the region [e.g. 51,84].
It was possible, however, to re-run the tests with the stated hypothesis that metrics from the
TB1 specimens are statistically significantly larger than the available extant/Holocene data. In
the case of this 1-tailed test, toothrow length of SF40 was found to be statistically significantly
larger than the Holocene dataset; with $p = 0.047$, however, this result is marginal (**table 3**).
Conversely, premolar row length in SF40 was not found to be statistically significantly larger
than the available comparative data. For SF42, length of p4 was not found to be statistically
significantly larger the comparative data set. Finally, canine width in SF43 was not found to
be statistically significantly larger than comparative data (**table 3**). The dimensions of the *H.*
*inermis* specimen from Lang Trang (PIN 579/20; L = 11.5 mm; W = 8.1 mm [53]), however,
suggest that the fourth premolar is relatively large in comparison to data from Holocene
animals. Length was found to be statistically significantly different from Holocene
comparatives and when the Upper Pleistocene specimens from TB1 are included (**table 3**).
While the p4 figured in [53] is worn, which can potentially affect measurements of teeth to a
degree [84], these data suggest that the specimen from Lang Trang reflects an earlier, larger-
toothed and/or larger-bodied form.

In summary, morphological and metric characters indicate that the TB1 specimens are
attributable to *Hydropotes*. There are no compelling grounds to suggest a larger, Pleistocene
subspecies and no reason (geographical or otherwise) for proposing a separate species. The
specimens from TB1 are referred to *H. inermis*. The identified remains derive from at least two
individuals. The size of the canine fragments indicate the finds include male animals [cf. 62].

5. Discussion

The remains of water deer from TB1 confirms the prehistoric presence of the species in
northern Vietnam. The specimens, dating to between 13,000- and 16,000-years BP, are valuable
contributions to the sparse Pleistocene record for the species and are, at present, the most
recent records in Vietnam. These new finds, however, raise questions.

Firstly, what were climatic and environmental conditions around TB1 between 13,000-
and 16,000-years BP? This was a period of significant global climatic instability | This is

326 also an important question as Upper Pleistocene zooarchaeological evidence from cave sites
within Trảng An indicate a focus on the exploitation of forest-adapted taxa from the karst
[56,86]. The position of TB1 in a “satellite” hill overlooking the plain to the west of the karst,
however, is likely to be pertinent (**figure 2**).

Observations of extant animals indicate that, while interlinked forest patches are a
significant factor that can mediate localised abundance [24], water deer tend to be a lowland
species with a preference for more open, marginal and riparian habitats. These habitats
include reed-beds and tall, damp and undisturbed grasslands, with forbs and woody plants
such as species of Asteraceae, Leguminosae and Fagaceae as favoured food plants [87-89].
Were similar habitats present on the plain outside the cave, or were the Upper Pleistocene
water deer of TB1 living in different conditions? Palaeoenvironmental data for the last
deglaciation in Northern Vietnam are sparse although records clearly show that rapid climatic
fluctuations were experienced during this period [90]. Inferences for the period between
16,000 to 13,000 years BP for the Trảng An karst and the intervening plains, can be synthesised
from a combination of proxies at various spatial scales.

[revised manuscript text omitted]

Currently, only two other sites in Vietnam, Mai Da Dieu and Lang Trang, have yielded
possible evidence of water deer; specimens from the former remain unconfirmed. Both these
sites are in northern Vietnam and are Upper Pleistocene in age. It is not possible to determine
any directly associated radiocarbon dates with the unconfirmed Mai Da Dieu material, but
dates given in the 2011 publication [52] suggest a post-LGM context and a broadly similar age
to the TB1 specimens. The tooth reported from Lang Trang is older and currently dated to
between 80,000-100,000 years BP [53]. The Pleistocene distribution of a species is not typically
perceived as a justification for designating it as “native” within the modern area. Firstly,
potential source populations for reintroductions may derive from a different genetic
population of the species. In this case, however, concerns about differing genetic lineages may
perhaps be allayed by the questionable subspecific status of extant water deer. Secondly, given
the climatic and environmental change that occurred during the Pleistocene-Holocene
transition, Pleistocene populations may have been adapted to and reliant upon different
conditions than the Holocene [15].

Water deer are today regarded as a temperate species (e.g. [89, 95]) and extant and
Quaternary records of water deer do tend towards temperate latitudes (**figure 1**). As we have
seen, available proxies indicate that the Upper Pleistocene landscapes of Trảng An reflected
cooler conditions than the sub-tropical climate of the Holocene. Regional
palaeoenvironmental proxies also indicate that a drier and cooler climate at the beginning of
the Holocene, following the Younger Dryas stadial, which saw reduced temperature and
precipitation approximately 12,900 to 11,700 years BP. While water deer may have persisted
into the early Holocene, increasing temperature and humidity and the establishment of
subtropical climate and environmental change [96] may have precluded the persistence of
populations of water deer in Northern Vietnam.

This argument, at least in part, would be an artefact of the shifting baseline syndrome.
The historical distribution of water deer in China, and the recent finds in Taiwan [30, 32]
present a significant challenge to this simplistic climatic and environmental narrative.
Historical records place water deer at sub-tropical latitudes in China until recently, in the 20th
century. In addition to the identified archaeological material, historical documents indicate
that water deer were, until relatively recently, common and widespread in Taiwan until
extirpation in the 19th century [32].

In practical terms, only confirmed and reliably dated specimens would provide
unequivocal evidence of populations surviving into the Holocene. A search for potential
specimens from Holocene age archaeological sites in Vietnam, either in archives or during
future investigations would be worthwhile, as would a review of potential documentary
evidence. A comprehensive assessment would also require other factors such as hunting by
humans, or competitive exclusion by other species to be considered, but there are currently
no clear climatic or environmental factors that would have been a definitive barrier to the
survival of water deer into the Holocene in northern Vietnam.

**6. Conclusion**

Newly identified fossils from the archaeological cave site of Hang Thung Binh 1 in the Tràng
An World Heritage Landscape Complex Site confirm the presence of water deer, *H. inermis*,
in northern Vietnam in the Upper Pleistocene. The new specimens are further evidence of a
wider Quaternary distribution for these Vulnerable, globally declining cervids, are valuable
additions to a sparse Pleistocene fossil record and confirm water deer as a component of the
Upper Pleistocene fauna of northern Vietnam.

Current archaeological and palaeontological evidence for the presence of water deer
in the country is extremely sparse and restricted to the Upper Pleistocene. While water deer
are today associated with temperate latitudes, a brief survey of the known historical
distribution and recent archaeological evidence indicates that the species has been capable of
surviving in sub-tropical climates and habitats. The possibility that water deer survived into
the Holocene in northern Vietnam is a hypothesis to be tested.

**Acknowledgements**

We thank the Tràng An Management Board, the Ninh Binh Provincial People's Committee
and Xuan Truong Construction Enterprise for ongoing support for research in Tràng An. Dr
Alex Wilshaw cleaned and conserved the archaeological material in the field. We thank all
our colleagues from the SUNDASIA project field teams, Ninh Binh People's Museum,
Vietnamese Institute of Archaeology (Hanoi) and Tràng An Management Board. CMS would
like to acknowledge the assistance and support of Eileen Westwig, Mark Carnall, Eliza
Howlett and the Oxford University Museum of Natural History. The study benefited from
access to the comparative museum collections, as well as those of the American Museum of
Natural History and Natural History Museum, UK.

**Data accessibility.** The datasets supporting this article have been uploaded as electronic
supplementary material.

**Author contributions.** CMS and RR conceived the study; CMS conducted analyses and
identification and drafted the manuscript with contributions from all authors; CMS, RH and
BU conducted the review of Quaternary records of *Hydropotes inermis*; CMS and BU
performed statistical tests; TK translated German language source material; CMS, BU, RR,
TK, carried out excavation, site analysis and recording; SO and NTMH produced the
paleoenvironmental synthesis. All authors revised the manuscript and gave final approval
for publication.

**Competing interests.** We declare we have no competing interests.

[revised manuscript text omitted]

76. Agelink van Rentergem JA, Huizenga HM. 2016 E-clip, Multivariate and Univariate Normative
Comparisons: eclip.shinyapps.io/NormativeComparisons/
77. Hammer Ø, Harper DAT, Ryan PD. 2001 PAST: Paleontological Statistics Software Package for
Education and Data Analysis. *Palaeontol. Electron.* **4**, 1-9.
78. Anders U, von Koenigswald W, Ruf I, Smith BH. 2011 Generalized individual dental age stages
for fossil and extant placental mammals. *PalZ* **85**, 321-39. (doi: 10.1007/s12542-011-0101-5)
79. Seo H, Kim J, Seomun H, Hwang JJ, Jeong HG, Kim JY, Kim HJ, Cho SW. 2017 Eruption of
posterior teeth in the maxilla and mandible for age determination of water deer. *Arch. Oral Biol.* **73**,
237-242. (doi: 10.1016/j.archoralbio.2016.10.020)
80. Rüttimeyer, L. 1881 *Beiträge zu einer natürlichen Geschichte der Hirsche*. Zürcher und Furrer. (In
German)
81. Groves C. 2016 Systematics of the Artiodactyla of China in the 21st century. *Zool. Res.* **37**, 119-125.
82. Wang Y, Harris R. 2015 *Moschus berezovskii*. The IUCN Red List of Threatened Species 2015:
e.T13894A103431781. <http://dx.doi.org/10.2305/IUCN.UK.2015-4.RLTS.T13894A61976926.en>
83. Leslie Jr. DM, Lee DN, Dolman RW. 2013 *Elaphodus cephalophus* (Artiodactyla: Cervidae). *Mamm.*
*Species* **45**, 80-91. (doi:10.1644/904.1)
84. Hooijer DA. 1958 Fossil Bovidae from the Malay Archipelago and the Punjab. *Zoologische*
*Verhandelingen*, **38**, 1-112.
85. Dykoski CA, Edwards RL, Cheng H, Yuan D, Cai Y, Zhang M, Lin Y, Qing J, An Z, Revenaugh J.
2005 A high-resolution, absolute-dated Holocene and deglacial Asian monsoon record from
Dongge Cave, China. *Earth Planet. Sci. Lett.* **233**, 71-86. (doi: 10.1016/j.epsl.2005.01.036)
86. Rabett R, Appleby J, Blyth A, Farr L, Gallou A, Giffiths T, Hawkes J, Marcus D, Marlow L, Morley
685 M, Tân NC, Son NV, Penkman K, Reynolds T, Stimpson, C, Szabó K. 2011 Inland shell midden
site-formation: investigation into a late Pleistocene to early Holocene midden from Trảng An,
northern Vietnam. *Quat. Int.* **239**, 153-169. (doi: 10.1016/j.quaint.2010.01.025)
87. Guo G, Zhang E. 2005 Diet of the Chinese water deer (*Hydropotes inermis*) in Zhoushan
Archipelago, China. *Acta Theriol. Sin.* **25**, 122-130.
88. Kim BJ, Lee NS, Lee SD. 2011 Feeding diets of the Korean water deer (*Hydropotes inermis*
*argyropus*) based on a 202 bp rbcL sequence analysis. *Conserv. Genet.* **12**, 851-856.
(doi:10.1007/s10592-011-0192-2)
89. Kim BJ, Oh DH, Chun SH, Lee SD. 2011 Distribution, density, and habitat use of the Korean water
deer (*Hydropotes inermis argyropus*) in Korea. *Landsc. Ecol. Eng.* **7**, 291-297. (doi:10.1007/s11355-010-
0127-y)
90. McAdams C, Morley MW, Fu X, Kandyba AV, Derevianko AP, Nguyen DT, Doi NG, Roberts RG.
2020 The Pleistocene geoarchaeology and geochronology of Con Moong Cave, North Vietnam: site
formation processes and hominin activity in the humid tropics. *Geoarchaeology* **35**, 72-97.
91. Huong NTM, Hai PV. 2009 Pollen and spore recorded at Con Moong Site (Thanh Hóa). *Vietnam*
*Archaeology* **4**, 24-31.
92. Tanabe S, Hori K, Saito Y, Haruyama S, Doanh LQ, Sato Y, Hiraide S. 2003 Sedimentary facies
and radiocarbon dates of the Nam Dinh-1 core from the Song Hong (Red River) delta, Vietnam. *J.*
*Asian Earth Sci.* **21**, 503-513. (doi: 10.1016/S1367-9120(02)00082-2)
93. Duong NT, Lieu NTH, Cuc NTT, Saito Y, Huong NTM, Phuong NTM, Thuy AT. 2020 Holocene
paleoshoreline changes of the Red River Delta, Vietnam. *Rev. Palaeobot. Palynol.* **278**, 104235. (doi:
10.1016/j.revpalbo.2020.104235)
94. Li Z, Saito Y, Matsumoto E, Wang Y, Haruyama S, Hori K, Doanh LQ. 2006 Palynological record
of climate change during the last deglaciation from the Song Hong (Red River) delta, Vietnam.
*Palaeogeogr. Palaeoclimatol. Palaeoecol.* **235**, 406-430. (doi:10.1016/j.palaeo.2005.11.023)
95. Rössner GE, Costeur L, Scheyer TM. 2020 Antiquity and fundamental processes of the antler cycle
in Cervidae (Mammalia). *The Science of Nature* **108**, 3 (doi:10.1101/2020.07.17.208587).

96. Dodson J, Li J, Lu F, Zhang W, Yan H, Cao S. 2019 A Late Pleistocene and Holocene vegetation and environmental record from Shuangchi Maar, Hainan Province, South China. *Palaeogeogr. Palaeoclimatol. Palaeoecol.* **523**, 89-96. (doi: 10.1016/j.palaeo.2019.03.026)

List of Figures

Figure 1. East Asia showing location of Tràng An (red triangle: this study) and distributions of *Hydropotes inermis*: extant range (yellow shading [29]); range in 20th century and estimated maximum westward extent of historical range (orange shading and dashed yellow line, respectively [30]); Holocene records (white circles [11, 32-36]); Upper Pleistocene records (black diamonds [38-43]) and Middle Pleistocene records (black squares [38, 44-48]). Data in table S1. Map (ESRI Satellite base map, EPSG: 3857-WGS 84 Pseudo-Mercator projection) produced in QGIS [49].

Figure 2. Hang Thung Binh 1. (a) Tràng An karst in plan, with location of Hang Thung Binh 1 (TB1 – red triangle) (b) looking north across the coastal plain, towards isolated hill containing TB1 (red triangle) with the Tràng An karst to the east and pagoda complex in the background (original photograph: TK) (c) TB1 in plan showing location of trenches and section D (d) representative west-facing section, showing calibrated radiocarbon dates (cal. BP) and find levels of *Hydropotes inermis* specimens SF40, SF42, SF43 and SF44. Levels are metres above sea level.

Figure 3. Upper Pleistocene specimens of *Hydropotes inermis* recovered from the Hang Thung Binh 1 archaeological cave site in the Tràng An World Heritage Area, Ninh Binh, Northern Vietnam. SF40 right mandible with tooththrow, p2-m3, (a) labial side (b) lingual side and (c) occlusal surface of teeth. SF42 right mandibular fragment with p4 and m1 in situ, (d) labial side (e) lingual side and (f) occlusal surface of teeth. SF43 left maxillary canine (g) lateral and (h) medial aspects. SF44 left maxillary canine tip (i) lateral and (j) medial aspects. Scale = 20 mm.

Figure 4. (a) Tooththrow lengths (p2-m3) and p4 lengths of SF40 and SF42 shown with equivalent data from three taxa, *Hydropotes inermis*, *Moschus moschiferus* and *Elaphodus cephalophus* (tables S3-S5) (b) Details of occlusal surface of teeth of the lower jaw of comparative taxa showing characters referred to in the text. All specimens are at individual dental age stage – IDAS 3 [78]. Specimen details: *Muntiacus muntjac* (OUMNH.ZC-20196); *Moschus moschiferus* (OUMNH.ZC-2894) *Hydropotes inermis* (redrawn from [79]) *Elaphodus cephalophus* (BMNH 92.7.13.1).

List of Tables

Table 1 Provenance and description of identified specimens of *Hydropotes inermis* from the Hang Thung Binh 1 archaeological cave site in the Tràng An World Heritage Landscape Complex Site, Ninh Binh, Northern Vietnam.

Table 2 Measurements of three *Hydropotes inermis* specimens from the archaeological cave site of Hang Thung Binh (TB1) in the Tràng An World Heritage Area, Ninh Binh, Northern Vietnam. All measurements are in mm. C – maxillary canine; p – mandibular premolar; m – mandibular molar; L – length (anterior-posterior); W – width (labial-lingual); p2-p4 – mandibular premolar row length; m1-m3 – mandibular molar row length; p2-m3 – mandibular tooththrow length.

Table 3 Summary of statistical output for tests of normality (Shapiro Wilk W) and univariate normative comparisons (modified *t*-test) of specimens from Tràng An and Lang Trang with comparative data (tables S3-S5). Significant results are shown in bold. Taxon data sets: ¹Holocene; ²Pleistocene and Holocene; ³Upper Pleistocene

Figure 1. East Asia showing location of Trảng An (red triangle: this study) and distributions of *Hydropotes inermis*: extant range (yellow shading [29]); range in 20th century and estimated maximum westward extent of historical range (orange shading and dashed yellow line, respectively [30]); Holocene records (white circles [11, 32-36]); Upper Pleistocene records (black diamonds [38-43]) and Middle Pleistocene records (black squares [38, 44-48]). Data in table S1. Map (ESRI Satellite base map, EPSG: 3857-WGS 84 Pseudo-Mercator projection) produced in QGIS [49].

160x113mm (200 x 200 DPI)

Figure 2. Hang Thung Binh 1. (a) Tràng An karst in plan, with location of Hang Thung Binh 1 (TB1 – red triangle) (b) looking north across the coastal plain, towards isolated hill containing TB1 (red triangle) with the Tràng An karst to the east and pagoda complex in the background (original photograph: TK) (c) TB1 in plan showing location of trenches and section D (d) representative west-facing section, showing calibrated radiocarbon dates (cal. BP) and find levels of *Hydropotes inermis* specimens SF40, SF42, SF43 and SF44. Levels are metres above sea level.

248x240mm (200 x 200 DPI)

Figure 3. Upper Pleistocene specimens of *Hydropotes inermis* recovered from the Hang Thung Binh 1 archaeological cave site in the Tràng An World Heritage Area, Ninh Binh, Northern Vietnam. SF40 right mandible with tooththrow, p2-m3, (a) labial side (b) lingual side and (c) occlusal surface of teeth. SF42 right mandibular fragment with p4 and m1 in situ, (d) labial side I lingual side and (f) occlusal surface of teeth. SF43 left maxillary canine (g) lateral and (h) medial aspects. SF44 left maxillary canine tip (i) lateral and (j) medial aspects. Scale = 20 mm.

162x107mm (300 x 300 DPI)

Figure 4. (a) Toothrow lengths (p2-m3) and p4 lengths of SF40 and SF42 shown with equivalent data from three taxa, *Hydropotes inermis*, *Moschus moschiferus* and *Elaphodus cephalophus* (tables S3-S5) (b) Details of occlusal surface of teeth of the lower jaw of comparative taxa showing characters referred to in the text.

All specimens are at individual dental age stage – IDAS 3 [78]. Specimen details: *Muntiacus muntjak* (OUMNH.ZC-20196); *Moschus moschiferus* (OUMNH.ZC-2894) *Hydropotes inermis* (redrawn from [79]) *Elaphodus cephalophus* (BMNH 92.7.13.1).

139x164mm (200 x 200 DPI)

Table 1 Provenance and description of identified specimens of *Hydropotes inermis* from the Hang Thung Binh 1 archaeological cave site in the Tràng An World Heritage Landscape Complex Site, Ninh Binh, Northern Vietnam.

Site	Trench	Grid sq.	Context	SF no.	Description
TB1	2	TR2EE	F908.2	40	right mandibular body and complete toothrow p2-m3
TB1	2	MS-E	F912	42	right mandible fragment with p4 and m1
TB1	2	MS-E	F912	43	left maxillary canine with root and crown.
TB1	2	MS-E	F912	44	fragment of left maxillary canine - tip of crown

Table 2 Measurements of three *Hydropotes inermis* specimens from the archaeological cave site of Hang Thung Binh (TB1) in the Tràng An World Heritage Area, Ninh Binh, Northern Vietnam. All measurements are in mm. C – maxillary canine; p – mandibular premolar; m – mandibular molar; L – length (anterior-posterior); W – width (labial-lingual); p2-p4 – mandibular premolar row length; m1-m3 – mandibular molar row length; p2-m3 – mandibular tooththrow length.

tooth/toothrow	dimension	SF40	SF42	SF43
C	W	/	/	10.98
p2	L	6.56	/	/
	W	3.19	/	/
p3	L	8.2	/	/
	W	5.36	/	/
p4	L	8.64	9.44	/
	W	5.86	6.24	/
m1	L	9.82	9.8	/
	W	7.2	7.38	/
m2	L	11	/	/
	W	7.11	/	/
m3	L	13.8	/	/
	W	6.68	/	/
p2-p4	L	24.2	/	/
m1-m3	L	34.45	/	/
p2-m3	L	57.96	/	/

Table 3 Summary of statistical output for tests of normality (Shapiro Wilk W) and univariate normative comparisons (modified *t*-test) of specimens from Tràng An and Lang Trang with comparative data (tables S3-S5). Significant results are shown in bold. Taxon data sets: ¹Holocene; ²Pleistocene and Holocene; ³Upper Pleistocene

Spec	Measure.	Taxon	n	W	p (norm.)	hyp	sig	diff	mod. t	p
SF40	L p2-m3	H. inermis ³	28	0.9398	0.1093	2-tailed	N	1.778	1.747	0.084
SF40	L p2-m3	E. cephalophus ¹	17	0.9215	0.1566	1-tailed (smaller)	Y	-1.972	-1.916	0.034
SF40	L p2-m3	M. moschiferus ²	10	0.9287	0.4353	1-tailed (larger)	Y	5.025	4.791	<0.001
SF40	L p2-p4	H. inermis ¹	24	0.9297	0.09588	2-tailed	N	0.78	0.765	0.454
SF40	L p4	E. cephalophus ¹	7	0.9312	0.5611	1-tailed (smaller)	Y	-3.539	-3.31	0.008
SF40	L p4	H. inermis ¹	7	0.9457	0.6902	2-tailed	N	0.261	0.244	0.826
SF42	L p4	H. inermis ¹	7	0.9457	0.6902	2-tailed	N	1.384	1.295	0.23
SF42	L p4	E. cephalophus ¹	7	0.9312	0.5611	1-tailed (smaller)	N	-0.545	-0.51	0.294
SF42	L p4	M. moschiferus ²	7	0.9261	0.518	1-tailed (larger)	Y	6.326	5.917	<0.001
SF43	Cw	H. inermis ¹	8	0.9184	0.4174	2-tailed	N	-0.061	-0.058	0.943
SF40	L p2-m3	H. inermis ¹	28	0.9398	0.1093	1-tailed (larger)	Y	1.778	1.747	0.047
SF40	L p2-p4	H. inermis ¹	24	0.9297	0.09588	1-tailed	N	0.78	0.765	0.224
SF42	L p4	H. inermis ¹	7	0.9457	0.6902	1-tailed	N	1.384	1.295	0.118
SF43	Cw	H. inermis ¹	8	0.9184	0.4174	2-tailed	N	-0.061	-0.058	> 0.9
SF40	L m3	E. cephalophus ³	24	0.9499	0.2696	1-tailed	Y	-4.038	-3.957	<0.001
PIN 5792 /20	L p4	H. inermis ¹	7	0.9457	0.6902	2-tailed	Y	4.276	4	<0.001
PIN 5792 /20	L p4	H. inermis ²	9	0.9318	0.4983	2-tailed	Y	4.177	3.963	0.01

Appendix B

Paleontological Society

The Morphology of the Lower Fourth Premolar as a Taxonomic Character in the Ruminantia (Mammalia; Artiodactyla), and the Systematic Position of Triceromeryx

Author(s): Christine M. Janis and Adrian Lister

Source: *Journal of Paleontology*, Vol. 59, No. 2 (Mar., 1985), pp. 405-410

Published by: SEPM Society for Sedimentary Geology

Stable URL: <http://www.jstor.org/stable/1305034>

Accessed: 21/01/2009 07:33

Your use of the JSTOR archive indicates your acceptance of JSTOR's Terms and Conditions of Use, available at <http://www.jstor.org/page/info/about/policies/terms.jsp>. JSTOR's Terms and Conditions of Use provides, in part, that unless you have obtained prior permission, you may not download an entire issue of a journal or multiple copies of articles, and you may use content in the JSTOR archive only for your personal, non-commercial use.

Please contact the publisher regarding any further use of this work. Publisher contact information may be obtained at <http://www.jstor.org/action/showPublisher?publisherCode=sepm>.

Each copy of any part of a JSTOR transmission must contain the same copyright notice that appears on the screen or printed page of such transmission.

JSTOR is a not-for-profit organization founded in 1995 to build trusted digital archives for scholarship. We work with the scholarly community to preserve their work and the materials they rely upon, and to build a common research platform that promotes the discovery and use of these resources. For more information about JSTOR, please contact support@jstor.org.

Paleontological Society and *SEPM Society for Sedimentary Geology* are collaborating with JSTOR to digitize, preserve and extend access to *Journal of Paleontology*.

<http://www.jstor.org>

THE MORPHOLOGY OF THE LOWER FOURTH PREMOLAR AS A
TAXONOMIC CHARACTER IN THE RUMINANTIA
(MAMMALIA; ARTIODACTYLA), AND THE
SYSTEMATIC POSITION OF *TRICEROMERYX*

CHRISTINE M. JANIS¹ AND ADRIAN LISTER

Department of Zoology, University of Cambridge, Cambridge, England

ABSTRACT—The morphology of the lower fourth premolar has been used as a taxonomic indicator in the pecoran fossil record, for example to separate cervid species, and to differentiate giraffids from cervoids. We demonstrate considerable inter- and intraspecific variation in cervoid and bovid P_4 morphology, and show in particular that the typical giraffid type of P_4 , with suppression of the central connection between labial and lingual walls, appears among living and fossil cervoids, and very exceptionally among bovinds. This character of the premolar dentition has been used to unite the lower Miocene pecoran genus *Triceromeryx*, known from a single incomplete individual from Spain, with the Giraffidae, although it was originally united with the Dromomerycidae on the basis of the possession of an occipital horn. We conclude that the known material of *Triceromeryx* is insufficient to assign this genus firmly to any known pecoran family, and that it should remain for the time being as Pecora incertae sedis.

INTRODUCTION

THE primitive dental condition for the suborder Ruminantia (Mammalia, Artiodactyla) is for simple sectorial premolars, and this condition is seen today in living mouse-deer (infraorder Tragulina). However, in the infraorder Pecora the more derived condition of submolariform premolars is seen in the earliest and most primitive members (Webb and Taylor, 1980). Within different pecoran lineages the premolars have apparently become more molariform in independent fashions. Both the pattern of molarization, and the details of the manner in which these teeth have become complicated, have been given much significance in attempts to determine the taxonomic interrelationships within ruminant artiodactyls (e.g., Loomis, 1925; Hamilton, 1973).

Information gleaned from premolar morphology has been of particular importance in assigning fossil taxa to the appropriate lineage, and lower premolar morphology has been particularly useful as lower dentitions are more abundant in the fossil record than are upper ones. Firstly, several workers have used lower premolars as a basis for identifying giraffoids in the fossil record (e.g., Colbert, 1936; Churcher, 1970; Hamilton,

1978a). They pointed out that there is a characteristic difference in the formation of the anterior part of the lower fourth premolar (P_4) which distinguishes giraffes from other ruminants that molarize this premolar. Whereas in cervids and bovinds the lingual wall is connected to the labial wall by a transverse crest [the "central transverse crest" of Hamilton (1978a)] (Figure 1.2-1.6), in giraffes this crest is suppressed (Figure 1.7).

Secondly, the composition of the lingual wall itself, and in particular the degree to which the paraconid and metaconid contribute to it, has been used as a taxonomic character to separate genera and species within the Cervidae and Bovidae (e.g., Azzaroli, 1953; Gentry and Gentry, 1978).

It is our contention that the morphology of the anterior part of the lower fourth premolar is much more variable than has hitherto been realized, especially among cervoids. While a particular pattern may be the general condition for each of the various pecoran lineages, the amount of individual variation is such that this character cannot be used in isolation to assign a single fossil specimen to a particular lineage. This problem was foreseen to some extent by Hamilton (1978b) when he recognized that the living cervid genus *Rangifer* possesses a "giraffe" type of lower fourth premolar. The same is true also of *Alces*. These two genera in fact

¹ Present address: Division of Biology and Medicine, Brown University, Providence, RI 02912.

FIGURE 1—Pecoran lower fourth premolar patterns. Schematic. Left side shown.

have a greater degree of molarization of the premolar row than most other cervids. Our concern in this paper is to demonstrate the variability of P_4 form within and between other cervoid, and bovid, lineages, and to relate this to the assignment of the problematical pecoran *Triceromeryx pachecoi*, known from a single incomplete specimen from the lower Miocene of Spain (Crusafont Pairó, 1952).

MATERIALS AND METHODS

Populations of living and fossil pecorans were examined for variation in P_4 morphology from the following institutions: the British Museum (Natural History), London, England; the Naturhistorisches Museum, Mainz, Germany; the Forschungsinstitut Senckenberg, Frankfurt, Germany; the Frick Collection, American Museum of Natural History, New York, U.S.A.; the Museum of Comparative Zoology, Harvard University, U.S.A.; the National Museum of Kenya, Nairobi, Kenya; the Transvaal Museum, Pretoria, South Africa; the Cape Town Museum, South Africa.

RESULTS

The results for cervoid pecorans are summarized in Table 1; cf. Figure 1. Considerable variation in P_4 form is apparent between and even within species, both living and fossil. The anterior part of the lingual wall may be absent (Figure 1.2), or formed by the metaconid (Figure 1.3), or by the paraconid (Figure 1.4), and in some cases a typically "giraffid" condition is seen, with suppression of the central transverse crest (Figure 1.7). In some individuals of the type shown in Figure 1.3 and 1.4, the paraconid and metaconid may fuse, especially lower in the crown, to form a continuous structure, as in Figure 1.5. The same fusion may occur in the "giraffid" type; that shown in Figure 1.7 is in the fused state.

Particularly interesting variability is seen in the dromomerycid *Dromomeryx whitfordi* from the middle Miocene of North America, in which three forms of molarization (types 3, 4 and 7) occur within a single population. There is also evidence that the frequencies of the variants within a species can change markedly, as shown by the figures for two different populations of *Cervus elaphus*.

For the Bovidae, we have examined a total of about 400 specimens from 33 living species, covering all of the African and Indian tribes. The species studied are listed in Table 2. Considerable variation was again seen in the development of the metaconid and paraconid. The most frequent pattern among these bovid tribes is that of Figure 1.6, where the metaconid tends to fuse to the entoconid rather than to the paraconid. However, P_4 's of types 2–4 are also occasionally observed (see also Gentry, 1978, fig. 27.1). In addition, P_4 type 5, with paraconid and metaconid joined to form a continuous anterolingual wall even high in the crown, has been observed as a variant within certain species of the tribes Cephalophini, Reduncini, Alcelaphini, Tragelaphini and Bovini (see Table 2). We have not to date observed this condition in any specimen of bovids from the tribes Neotragini, Gazellini or Hippotragini, although our sample sizes for some species are small (see Table 2). The distribution of this form of P_4 amongst the bovid tribes revealed by these results broadly corresponds to that given by Gentry (1978). However its occurrence, even

TABLE 1—Cervoid specimens examined. Institutions: BMNH, British Museum (Natural History); AMNH, American Museum of Natural History; MCZ, Museum of Comparative Zoology; NHM, Naturhistorisches Museum; NMN, National Museum of Kenya, Nairobi; CTM, Cape Town Museum; TM, Transvaal Museum. 1 to 7: types of P_4 pattern as in Figure 1. 3(-5): showing type 3, or type 5 where it is clear that this is a fused or advanced-wear stage of 3. 4(-5): showing type 4 or type 5 where it is clear that this is a fused or advanced-wear stage of 4. 3/4/5: P_4 type 3, 4 or 5, indistinguishable because of advanced stage of wear. N: number of individuals examined. Percentages: percentages of specimens in each sample showing each type of P_4 . Note: in the *Capreolus* specimens scored as type 7, the central transverse crest varies from being absent to being weakly developed, but never joins the metaconid.

Species	Age and provenance of sample	Institution	1	2	3(-5)	4(-5)	3/4/5	6	7	N
Dicrocerus elegans	Early Miocene (Sansan), France	BMNH, London	83%	17%						6
Palaeomeryx bojani	Early Miocene (Sansan), France	BMNH, London	33%	67%						3
Dromomeryx whitfordi	Barstovian (Mid. Miocene), U.S.A.	Frick Coll., AMNH, New York			42%	8%	47%		3%	62
Cervus elaphus	Mid. Pleistocene, Mosbach, Germany	NHM, Mainz, Germany		8%	70%	7%	15%			60
Cervus elaphus	Early Holocene, Star Carr, England	BMNH, London		47%	20%	20%	13%			30
Capreolus capreolus	Recent, England	BMNH, London			5%		3%		91%	58
Muntiacus muntjac	Recent, S. E. Asia	BMNH, London		84%		4%	7%	5%		57
Moschus moschiferus	Recent, India	BMNH, London		3%	56%	3%	28%		9%	32
Odocoileus virginianus chiniquensis	Recent, Panama Canal, U.S.A.	MCZ, Harvard, U.S.A.			10%		68%		22%	50

in those tribes where it is found, is not constant either between or within species.

The highest frequency of P_4 type 5 is among the Tragelaphini. In one specimen of *Tragelaphus angasi* observed, there was also complete suppression of the central transverse crest to produce a quasi-giraffid morphology (Figure 1.7). This is the only bovid specimen observed to show this condition.

Our giraffid sample was not as extensive as that for cervoids and bovids, owing to the relative scarcity of these animals in museum collections. However, we have never observed variation from the giraffid form of P_4 (Figure 1.7) in any skull examined, amounting to 35 specimens of *Giraffa camelopardis*, plus several of *Okapia johnstoni*, from museums in Europe, Africa and North America.

DISCUSSION

The results presented here show, firstly, that in species-level cervoid taxonomy it is un-

wise to assume that the P_4 morphology found in a given population of a certain species is a constant, taxonomically valid feature of that species. This is evident from the intra- and interpopulation variation seen in *Cervus elaphus*. Thus, to take a single example, the poor development of the metaconid noted in small samples of middle Pleistocene *Megaceros* species from England (Azzaroli, 1953), need not be diagnostic. We have not assessed in detail the relative contributions of intra- and interspecific variation in bovid P_4 's, but our results suggest that caution is advisable in bovid species-level taxonomy also.

Secondly, on the level of distinguishing the bovid tribes, it would seem that the presence of paraconid-metaconid fusion in P_4 is not a reliable feature of any African or Indian tribe, although its absence may be constant in certain tribes.

Thirdly, since a typically "giraffid" type of P_4 morphology occurs as the standard form

TABLE 2—Bovid specimens examined. : occurrence within a species, but not necessarily in all individuals. Other symbols and abbreviations as in Table 1.

Tribe	Species	Institutions	N	Occurrence of P ₄ type 5	Occurrence of P ₄ type 7
Boselaphini	Boselaphus tragocamelus	BMNH, CTM	14	—	—
Tragelaphini	Tragelaphus scriptus	BMNH, CTM, TM	24	—	—
	Tragelaphus angasi	CTM	6	+	+
	Tragelaphus strepsiceros	CTM, TM	17	+	—
	Boocercus eurycerus	NMN	1	+	—
	Taurotragus oryx	NMN, CTM, TM	12	+	—
Cephalophini	Cephalophus caeruleus	BMNH	15	—	—
	Cephalophus monticola	NMN	6	—	—
	Sylvicapra grimmia	NMN, CTM	28	+	—
Bovini	Syncerus caffer	CTM, TM	11	+	—
Neotragini	Neotragus moschatus	NMN	7	—	—
	Ourebia ourebi	CTM	2	—	—
	Oreotragus oreotragus	CTM, TM	6	—	—
	Raphicerus melanotis	CTM	10	—	—
	Raphicerus campestris	CTM	17	—	—
	Madoqua kirkii	NMN	3	—	—
Reduncini	Redunca arundinum	BMNH, CTM, TM	32	—	—
	Redunca fulvorufula	CTM, TM	17	—	—
	Pelea capreolus	CTM, TM	6	+	—
	Kobus ellipsiprymnus	NMN, CTM, TM	15	+	—
Gazellini (Antilopini)	Gazella thomsoni	BMNH	30	—	—
	Litocranius walleri	BMNH, NMN	16	—	—
	Ammodorcas clarkei	BMNH	3	—	—
	Antidorcas marsupialis	CTM	10	—	—
Hippotragini	Hippotragus niger	CTM, TM	11	—	—
	Hippotragus equinus	TM	2	—	—
	Oryx gazella	CTM, TM	10	—	—
Alcelaphini (sensu Vrba)	Aepyceros melampus	BMNH, CTM, TM	17	+	—
	Beatragus hunteri	BMNH	6	—	—
	Alcelaphus buselaphus	NMN, CTM, TM	18	—	—
	Damaliscus lunatus	NMN, CTM, TM	12	—	—
	Connochaetes taurinus	NMN, CTM, TM	20	+	—

in some cervid genera (*Rangifer*, *Alces*), and as a frequent variant in a number of other cervid species and in *Dromomeryx* among the dromomerycids, and even as a rare variant in at least one species of Bovidae, we conclude that this character cannot be used in isolation to assign a single fossil specimen to the family Giraffidae, as has been the case with *Triceromeryx pachecoi* (Churcher, 1970; Hamilton, 1978a).

Triceromeryx was originally assigned to the Dromomerycidae on the basis of possessing a forked occipital horn in addition to simple supra-orbital horns (Crusafont-Pairó, 1952), as a simple occipital horn was also present in the dromomerycid subfamily Cranioicerotinae (Frick, 1937), despite the fact that *Triceromeryx* was found in Spain and all known dromomerycids were exclusively North American. The taxonomic affinities of the

dromomerycids are uncertain, and thus *Triceromeryx* has been given significance as a possible clue to their origin and affinities, notwithstanding the very limited amount of material assignable to this genus. Dromomerycids are often lumped together with the paleomerycids, the so-called "basal" Eurasian pecoran group (e.g., Stirton, 1944; Romer, 1966). However, whereas the Old World "paleomerycids" can be shown to be a polyphyletic assemblage, with most members assignable to the Giraffidae (Hamilton, 1978b) or the Moschidae (Webb and Taylor, 1980), it seems likely that the Dromomerycidae are a monophyletic group, radiating in North America from an immigrant Eurasian pecoran in the lower Miocene (see Janis, 1982). They are variously claimed to be cervoids (e.g., Frick, 1937) or giraffoids (e.g., Stirton, 1944; Crusafont Pairó, 1952) as they share

with giraffes the elongation of the occipital region and the possession of a simple, unbranching, nondeciduous ossicone that was apparently covered with skin rather than keratin (the condition in bovids). However, dromomerycids possess the derived cervoid characters of a closed metatarsal gully (see Leinders, 1979) and a double lachrymal orifice (see Leinders and Heintz, 1980), and do not share the giraffe autapomorphy of a bilobed lower canine (see Hamilton, 1978a). It seems likely that they are a cervoid family, and that their superficial resemblances to giraffes are convergences, resulting from a similar feeding behavior of high-level browsing (Janis, 1982).

Triceromeryx was assigned to the Giraffidae by both Churcher (1970) and Hamilton (1978a) primarily on the basis of the form of the P_4 . Both authors refer to the post-cranial material of *Triceromeryx* as "paleotragine-like," and Crusafont Pairó (1952) noted that various details of the limbs, especially of the carpus and radius, were more similar to primitive giraffids such as *Palaeotragus* than to cervids or *Palaeomeryx*. However, cervid limb morphology seems to represent a derived condition for pecorans (Leinders, 1979), and the taxonomic affinities of the genus *Palaeomeryx* are uncertain (Hamilton, 1978a, 1978b). In the absence of a detailed evaluation of which pecoran limb characteristics are primitive and which are derived, direct comparison of *Triceromeryx* with *Palaeotragus* in this manner is meaningless, as both may merely represent the retention of the shared primitive condition. (It is unfortunate that the distal end of the metatarsus of *Triceromeryx* is lacking, so it is impossible to determine whether or not it had a cervoid type of closed gully. Also lacking is the anterior end of the lower jaw, so it cannot be determined if it shared the giraffe character of a bilobed canine.)

It may well be the case that *Triceromeryx* is an aberrant giraffid, especially as no other Old World material assignable to dromomerycids has as yet been identified. However, as the only positive character linking *Triceromeryx* with giraffes is the form of the P_4 , and this type of P_4 can be seen among both cervids and dromomerycids, it seems to us that there is insufficient evidence from the

material available at present firmly to unite this animal with any known pecoran family.

The lower Miocene Old World ruminants are poorly known, and clear characters that would unite them with various pecoran or traguloid families are not as yet determined (though see Webb and Taylor, 1980, for hornless ruminants). The ubiquitous Miocene genus *Palaeomeryx*, originally assigned to the Giraffidae by Ginsburg and Heitz (1966) on the basis of associated ossicone material, has been shown by Hamilton (1978a, 1978b) to lack derived giraffid characters. He classified the genus as *Pecora incertae sedis*, along with certain other Miocene "palaeomerycids" such as *Propalaeoryx* and *Prolibytherium*. We suggest that until more material of *Triceromeryx* is discovered, and until a full review is undertaken of primitive and derived character states in Miocene pecorans, *Triceromeryx* would also be best considered as *Pecora incertae sedis* rather than assigned either to the Giraffidae or the Dromomerycidae.

ACKNOWLEDGMENTS

We would like to acknowledge the following people for access to collections in their care: J. Clutton-Brock (British Museum of Natural History), Q. Hendey (Cape Town Museum), J. Hooker (BMNH), L. Jacobs (National Museum of Kenya), P. Jenkins (BMNH), J. Kirsch (Museum of Comparative Zoology, Harvard), O. Neuffer (Naturhistorisches Museum, Mainz), G. Storch (Forschungsinstitut Senckenberg), R. Tedford (American Museum of Natural History) and E. Vrba (Transvaal Museum). C. M. J. acknowledges useful conversations with A. Gentry and J. J. M. Leinders, and thanks Newnham College, Cambridge, England, for travel funds to visit the U.S.A. and for the Phyllis and Eileen Gibbs Travelling Fellowship which enabled her to visit museums in Africa. A. L. was funded by Gonville and Caius College, Cambridge, and the German Academic Exchange Service.

REFERENCES

- AZZAROLI, A. 1953. The deer of the Weybourne Crag and Forest Bed of Norfolk. *Bulletin of the British Museum (Natural History)*, London (A., Geology), 2:3-96.

- CHURCHER, C. S. 1970. Two new upper Miocene giraffids from Fort Ternan, Kenya, East Africa. *Palaeotragus primaevus* (n.sp.) and *Samotherium africanus* (n.sp.), p. 1–336. In L. S. B. Leakey and R. J. G. Savage, Fossil Vertebrates of Africa, Vol. 2. Academic Press, London and New York.
- COLBERT, E. H. 1936. *Palaeotragus* in the Tung Gur Formation of Mongolia. American Museum Novitates, 874, 17 p.
- CRUSAFONT PAIRÓ, M. 1952. Los jiráfidos fósiles de España. Memorias y Comunicaciones del Instituto Geológico, Barcelona, 8:1–239.
- FRICK, C. 1937. Horned ruminants of North America. Bulletin of the American Museum of Natural History, New York, 69, 669 p.
- GENTRY, A. W. 1978. Bovidae, p. 540–572. In V. J. Maglio and H. B. S. Cooke, Evolution of African Mammals. Harvard University Press, Cambridge.
- and A. GENTRY. 1978. Fossil Bovidae (Mammalia) of Olduvai Gorge, Tanzania. Part I. Bulletin of the British Museum (Natural History), London (A., Geology), 29:289–446.
- GINSBURG, L. and E. HEINTZ. 1966. Sur les affinités du genre *Palaeomeryx*. Compte Rendu Hebdomadaire des Séances de l'Académie des Sciences, Paris, 262:979–982.
- HAMILTON, W. R. 1973. The lower Miocene ruminants of Gebel Zelten, Libya. Bulletin of the British Museum (Natural History), London (A. Geology), 21:76–150.
- . 1978a. Fossil giraffes from the Miocene of Africa and a revision of the phylogeny of the Giraffoidea. Philosophical Transactions of the Royal Society of London, Series B, 283:165–229.
- . 1978b. Cervidae and Palaeomerycidae, p. 496–508. In V. J. Maglio and H. B. S. Cooke, Evolution of African Mammals. Harvard University Press, Cambridge.
- JANIS, C. 1982. Evolution of horns in ungulates: ecology and palaeoecology. Biological Reviews of the Cambridge Philosophical Society, 57:261–318.
- LEINDERS, J. J. M. 1979. On the osteology and function of the digits of some ruminants, and their bearing on taxonomy. Zeitschrift für Säugetierkunde, 43:42–49.
- and E. HEINTZ. 1980. The configuration of the lacrimal orifice in pecorans and tragulids (Artiodactyla; Mammalia) and its significance for the distinction between Bovidae and Cervidae. Beaufortia, 30:155–160.
- LOOMIS, F. B. 1925. Dentition of artiodactyls. Bulletin of the Geological Society of America, 36:583–604.
- ROMER, A. S. 1966. Vertebrate Paleontology (3rd edition). University of Chicago Press, Chicago.
- STIRTON, R. A. 1944. Comments on the relationship of the cervoid family Palaeomerycidae. American Journal of Science, 242:633–655.
- VRBA, E. 1979. Phylogenetic analysis and classification of fossil and recent Alcelaphini Mammalia: Bovidae. Biological Journal of the Linnean Society, 11:207–228.
- WEBB, S. D. and B. E. TAYLOR. 1980. The phylogeny of hornless ruminants and a description of the cranium of *Archaeomeryx*. Bulletin of the American Museum of Natural History, New York, 167:121–157.

MANUSCRIPT RECEIVED JUNE 8, 1982

REVISED MANUSCRIPT RECEIVED JUNE 21, 1983

Appendix C**ROYAL SOCIETY
OPEN SCIENCE****Confirmed archaeological evidence of water deer in
Vietnam: relics of the Pleistocene or a shifting baseline?**

Journal:	Royal Society Open Science
Manuscript ID	RSOS-210529
Article Type:	Research
Date Submitted by the Author:	28-Mar-2021
Complete List of Authors:	Stimpson, Christopher; Queen's University Belfast School of Natural And Built Environment; Oxford University Museum of Natural History O'Donnell, Shawn; Queen's University Belfast, School of Natural and Built Environment Mai Huong, Nguyen Thi; Vietnam Academy of Social Sciences, Institute of Archaeology Holmes, Rachael; University of Leicester, School of Geography, Geology and the Environment Utting, Benjamin; University of Cambridge, Archaeology Kahlert, Thorsten; Queen's University Belfast Rabett, Ryan; Queen's University Belfast
Subject:	taxonomy and systematics < BIOLOGY, palaeontology < BIOLOGY, environmental science < BIOLOGY
Keywords:	water deer, Hydropotes inermis, zooarchaeology, Vietnam, Pleistocene
Subject Category:	Ecology, Conservation, and Global Change Biology

Author-supplied statements

Relevant information will appear here if provided.

Ethics

Does your article include research that required ethical approval or permits?:

This article does not present research with ethical considerations

Statement (if applicable):

CUST_IF_YES_ETHICS :No data available.

Data

It is a condition of publication that data, code and materials supporting your paper are made publicly available. Does your paper present new data?:

Yes

Statement (if applicable):

The datasets supporting this article have been uploaded as electronic supplementary material.

Conflict of interest

I/We declare we have no competing interests

Statement (if applicable):

CUST_STATE_CONFLICT :No data available.

Authors' contributions

This paper has multiple authors and our individual contributions were as below

Statement (if applicable):

CMS and RR conceived the study; CMS conducted analyses and identification and drafted the manuscript with contributions from all authors; CMS, RH and BU conducted the review of Quaternary records of *Hydropotes inermis*; CMS and BU performed statistical tests; TK translated German language source material; CMS, BU, RR, TK, carried out excavation, site analysis and recording; SO and NTMH produced the paleoenvironmental synthesis. All authors revised the manuscript and gave final approval for publication.

[revised manuscript text omitted]

The dentition is adult but is unworn. Individual dental age score (IDAS) is early IDAS
 3 based on eruption and wear to the molars [78]. The p4 is "**molarised**" in that, in terms of
 gross morphology, this tooth comprises of two distinct lobes. The **anterior** lobe is broad,
 whereas the posterior lobe is compressed and **elongate** in the **labial-lingual** direction with
 well-developed posterolingual and posterolabial conids. The anterior and mesolingual conids
 are **joined**, forming a contiguous structure, which enclose an anterior fossette. The **lingual**
 edges of the metaconids of the molars are simple, with single shallow folds. Weakly
 developed anterior cingulids are present on the m1 and m2. A small, slender anterior
 ectostylid is present on the m3 (**figure 4**).

SF42 is a small fragment (maximum length of specimen = 32 mm) of the dorsal surface
 of the body of a right mandible (**figure 3**). The ventral side is broken away. Two complete,
 **unworn** teeth, p4 and m1, are preserved in situ and appear slightly larger and more robust
 than those in SF40. A portion of the alveolus for the m2 is also preserved. As in SF40, the p4
 is **also** "**molarised**" (**figure 4**), with an elongate posterior lobe and well-developed
 posterolingual and **posterolabial** conids. A weakly-developed anterior cingulid and a weakly-
 developed **elongate** (in the **labial-lingual** direction) ectostylid is present on the m1. The
 **lingual edge of the metaconids** lack complex folds or crenulations.

SF43 is a left maxillary canine (maximum length of specimen = 51.20 mm; **figure 3**).
 The specimen was recovered in two pieces and refitted. The closed root and majority of crown
 are preserved although the crown **is** chipped and broken at the tip (**figure 4**). The tooth curves
 posteriorly. The **lateral surface of the crown is concave**. The medial surface is relatively flat.

SF44 is the broken tip of a left **maxillary** canine (maximum length of specimen = 17.92
 232 mm; **figure 3**). As with SF43, SF44 is **convex** on the lateral side and flattened on the medial.

4.2 Diagnosis

Absolute size of specimens, tooth dimensions and morphology all indicate small ruminant:
 either a species of the Cervidae or Moschidae. The relatively long, posteriorly curved canine
 with a closed root also indicates the so-called “fanged” deer or musk deer.

The fourth lower premolars are present in both mandible fragments and are
 “molarised”. While Hooijer [51] describes one exceptional specimen of *M. muntjak vaginalis*
 (AMNH 43056), where the anterior and mesolingual conids meet (but are not fused), this
 character discounts *Muntiacus* [19, 51, 80] and indicates three candidate taxa: *Moschus* spp., *H.*
 *inermis* and *Elaphodus cephalophus* (figure 4; for *Moschus* spp., *M. moschiferus* is figured).

The smallest of the three taxa, the musk deer - *Moschus* spp. - are not true cervids and
 are classified in a separate family, Moschidae. Musk deer are currently distributed from the
 Himalayas to North East Asia, with up to seven extant species [81]. The smallest species, *M.*
 *berezovskii* (Flerov, 1928; extant body weight 6-9 kg), the “dwarf” or “forest musk deer”, ranges
 into South China and marginally into north Vietnam, where it has been reported in karst
 habitats. The much-reduced Vietnamese populations were, until recently, thought to be
 Siberian musk deer, *M. moschiferus* [82].

*Moschus* is indicated against here by the relatively simple lingual outline of the molars
 in SF40 and SF42. Specifically, “double-folded” lingual margins of the metaconids are key
 diagnostic characters of *Moschus* [63,80] and are absent in the archaeological material (figure
 4). Furthermore, anterior cingulids and mesostylids are relatively well developed in
 comparative specimens from the genus in comparison to the archaeological material.
 Conversely, posterolingual and posterolabial conids were well developed on the p4 in the
 available reference material.

Univariate comparison of dental measurements from extant and Pleistocene *M.*
 *moschiferus* with the archaeological specimens, found measures to be statistically significantly
 larger in the archaeological specimens (table 3). Although statistical comparison was not
 possible due to lack of a sufficient sample size, the width of canine SF43 was greater and the
 overall tooth approximately 30% larger than that of the comparative material for *Moschus*.

The largest of the three potential taxa, the tufted deer, *E. cephalophus*, (17-30kg) is
 currently found in southern China, with historical records from eastern Myanmar [83]. In
 terms of gross size and morphology, the mandibles and teeth of extant tufted deer appear
 larger and more robust than the TB1 material (figure 4). Relatively well developed ectostylids
 are apparent on the molars. If present, anterior cingula are very weak: they are more
 commonly absent in comparative material for *E. cephalophus*. The posterolabial and
 posterolingual conids on p4 are not as developed as in the archaeological specimens, but
 there is a relatively deep fold in the lingual wall between the anterior stylid and anterior conid,
 which is absent in the p4 of SF40 and SF42. These characters are also consistent in large
 Pleistocene specimens of the genus figured in [51]. The maxillary canine in *E. cephalophus*
 appears shorter, but much more robust than in SF43.

Toothrow length and p4 length of SF40 was found to be statistically significantly
 smaller than comparative data for extant tufted deer. No statistically significant difference
 was found in p4 length between SF42 and comparative data, however, and it is likely that
 individual tooth length may not be the most sensitive measure to discriminate between
 *Hydropotes* and *Elaphodus*. Comparisons of m3 lengths of Upper Pleistocene *E. cephalophus* [70]
 suggest that this metric is statistically significantly larger than SF40 (table 3).

In terms of morphology, the archaeological specimens most closely match the
available comparative material for water deer, *H. inermis*. The molarised p4 in *H. inermis* has
an elongate posterior lobe and well-developed posterolingual and posterolabial conids as in
both archaeological specimens (figure 4; see also [19] for the potential diagnostic significance
of posterolabial conids). Weakly developed anterior cingulids are present on the molars of
available comparative material. The lingual walls of the metaconids are relatively simple (i.e.,
lack complex folds), as in the archaeological specimens, a key character to distinguish
*Hydropotes* from *Moschus* [63,80].

While both archaeological mandibles and their teeth appear marginally larger and
more robust in comparison to extant reference material for *Hydropotes*, the dental metrics of
the archaeological specimens are within the ranges of equivalent available data for extant
specimens (figure 4). No statistically significant differences were found in lower tooth rows
lengths or premolar row lengths between SF40 and comparative data for *H. inermis*. No
differences were found between p4 lengths in SF40 and SF42 or available comparative data.
No significant difference was found in canine width between SF43 and comparative data for
*H. inermis* (table 3).

Confirmed Pleistocene records for *H. inermis* are scarce, as are Pleistocene age dental
data (with the exception of the p4 from Lang Trang, below). It was not possible to locate a
data set to investigate the possibility of larger Pleistocene body size in the region [e.g. 51,84].
It was possible, however, to re-run the tests with the stated hypothesis that metrics from the
TB1 specimens are statistically significantly larger than the available extant/Holocene data. In
the case of this 1-tailed test, toothrow length of SF40 was found to be statistically significantly
larger than the Holocene dataset; with $p = 0.047$, however, this result is marginal (table 3).
Conversely, premolar row length in SF40 was not found to be statistically significantly larger
than the available comparative data. For SF42, length of p4 was not found to be statistically
significantly larger the comparative data set. Finally, canine width in SF43 was not found to
be statistically significantly larger than comparative data (table 3). The dimensions of the *H.*
*inermis* specimen from Lang Trang (PIN 579/20; L = 11.5 mm; W = 8.1 mm [53]), however,
suggest that the fourth premolar is relatively large in comparison to data from Holocene
animals. Length was found to be statistically significantly different from Holocene
comparatives and when the Upper Pleistocene specimens from TB1 are included (table 3).
While the p4 figured in [53] is worn, which can potentially affect measurements of teeth to a
degree [84], these data suggest that the specimen from Lang Trang reflects an earlier, larger-
toothed and/or larger-bodied form.

In summary, morphological and metric characters indicate that the TB1 specimens are
attributable to *Hydropotes*. There are no compelling grounds to suggest a larger, Pleistocene
subspecies and no reason (geographical or otherwise) for proposing a separate species. The
specimens from TB1 are referred to *H. inermis*. The identified remains derive from at least two
individuals. The size of the canine fragments indicate the finds include male animals [cf. 62].

5. Discussion

The remains of water deer from TB1 confirms the prehistoric presence of the species in
northern Vietnam. The specimens, dating to between 13,000- and 16,000-years BP, are valuable
contributions to the sparse Pleistocene record for the species and are, at present, the most
recent records in Vietnam. These new finds, however, raise questions.

Firstly, what were climatic and environmental conditions around TB1 between 13,000-
and 16,000-years BP? This was a period of significant global climatic instability [85]. This is

also an important question as Upper Pleistocene zooarchaeological evidence from cave sites
within Trảng An indicate a focus on the exploitation of forest-adapted taxa from the karst
[56,86]. The position of TB1 in a “satellite” hill overlooking the plain to the west of the karst,
however, is likely to be pertinent (figure 2).

Observations of extant animals indicate that, while interlinked forest patches are a
significant factor that can mediate localised abundance [24], water deer tend to be a lowland
species with a preference for more open, marginal and riparian habitats. These habitats
include reed-beds and tall, damp and undisturbed grasslands, with forbs and woody plants
such as species of Asteraceae, Leguminosae and Fagaceae as favoured food plants [87-89].
Were similar habitats present on the plain outside the cave, or were the Upper Pleistocene
water deer of TB1 living in different conditions? Palaeoenvironmental data for the last
deglaciation in Northern Vietnam are sparse although records clearly show that rapid climatic
fluctuations were experienced during this period [90]. Inferences for the period between
16,000 to 13,000 years BP for the Trảng An karst and the intervening plains, can be synthesised
from a combination of proxies at various spatial scales.

[revised manuscript text omitted]

Currently, only two other sites in Vietnam, Mai Da Dieu and Lang Trang, have yielded
possible evidence of water deer; specimens from the former remain unconfirmed. Both these
sites are in northern Vietnam and are Upper Pleistocene in age. It is not possible to determine
any directly associated radiocarbon dates with the unconfirmed Mai Da Dieu material, but
dates given in the 2009 publication [52] suggest a post-LGM context and a broadly similar age
to the TB1 specimens. The tooth reported from Lang Trang is older and currently dated to
between 80,000-100,000 years BP [53]. The Pleistocene distribution of a species is not typically
perceived as a justification for designating it as “native” within the modern area. Firstly,
potential source populations for reintroductions may derive from a different genetic
population of the species. In this case, however, concerns about differing genetic lineages may
perhaps be allayed by the questionable subspecific status of extant water deer. Secondly, given
the climatic and environmental change that occurred during the Pleistocene-Holocene
transition, Pleistocene populations may have been adapted to and reliant upon different
conditions than the Holocene [15].

Water deer are today regarded as a temperate species (e.g. [89, 95]) and extant and
Quaternary records of water deer do tend towards temperate latitudes (**figure 1**). As we have
seen, available proxies indicate that the Upper Pleistocene landscapes of Trảng An reflected
cooler conditions than the sub-tropical climate of the Holocene. Regional
palaeoenvironmental proxies also indicate that a drier and cooler climate at the beginning of
the Holocene, following the Younger Dryas stadial, which saw reduced temperature and
precipitation approximately 12,900 to 11,700 years BP. While water deer may have persisted
into the early Holocene, increasing temperature and humidity and the establishment of
subtropical climate and environmental change [96] may have precluded the persistence of
populations of water deer in Northern Vietnam.

This argument, at least in part, would be an artefact of the shifting baseline syndrome.
The historical distribution of water deer in China, and the recent finds in Taiwan [30, 32]
present a significant challenge to this simplistic climatic and environmental narrative.
Historical records place water deer at sub-tropical latitudes in China until recently, in the 20th
century. In addition to the identified archaeological material, historical documents indicate
that water deer were, until relatively recently, common and widespread in Taiwan until
extirpation in the 19th century [32].

In practical terms, only confirmed and reliably dated specimens would provide
unequivocal evidence of populations surviving into the Holocene. A search for potential
specimens from Holocene age archaeological sites in Vietnam, either in archives or during
future investigations would be worthwhile, as would a review of potential documentary
evidence. A comprehensive assessment would also require other factors such as hunting by
humans, or competitive exclusion by other species to be considered, but there are currently
no clear climatic or environmental factors that would have been a definitive barrier to the
survival of water deer into the Holocene in northern Vietnam.

**6. Conclusion**

Newly identified fossils from the archaeological cave site of Hang Thung Binh 1 in the Tràng
An World Heritage Landscape Complex Site confirm the presence of water deer, *H. inermis*,
in northern Vietnam in the Upper Pleistocene. The new specimens are further evidence of a
wider Quaternary distribution for these Vulnerable, globally declining cervids, are valuable
additions to a sparse Pleistocene fossil record and confirm water deer as a component of the
Upper Pleistocene fauna of northern Vietnam.

Current archaeological and palaeontological evidence for the presence of water deer
in the country is extremely sparse and restricted to the Upper Pleistocene. While water deer
are today associated with temperate latitudes, a brief survey of the known historical
distribution and recent archaeological evidence indicates that the species has been capable of
surviving in sub-tropical climates and habitats. The possibility that water deer survived into
the Holocene in northern Vietnam is a hypothesis to be tested.

**Acknowledgements**

We thank the Tràng An Management Board, the Ninh Binh Provincial People's Committee
and Xuan Truong Construction Enterprise for ongoing support for research in Tràng An. Dr
Alex Wilshaw cleaned and conserved the archaeological material in the field. We thank all
our colleagues from the SUNDASIA project field teams, Ninh Binh People's Museum,
Vietnamese Institute of Archaeology (Hanoi) and Tràng An Management Board. CMS would
like to acknowledge the assistance and support of Eileen Westwig, Mark Carnall, Eliza
Howlett and the Oxford University Museum of Natural History. The study benefited from
access to the comparative museum collections, as well as those of the American Museum of
Natural History and Natural History Museum, UK.

**Data accessibility.** The datasets supporting this article have been uploaded as electronic
supplementary material.

**Author contributions.** CMS and RR conceived the study; CMS conducted analyses and
identification and drafted the manuscript with contributions from all authors; CMS, RH and
BU conducted the review of Quaternary records of *Hydropotes inermis*; CMS and BU
performed statistical tests; TK translated German language source material; CMS, BU, RR,
TK, carried out excavation, site analysis and recording; SO and NTMH produced the
paleoenvironmental synthesis. All authors revised the manuscript and gave final approval
for publication.

**Competing interests.** We declare we have no competing interests.

[revised manuscript text omitted]

76. Agelink van Rentergem JA, Huizenga HM. 2016 E-clip, Multivariate and Univariate Normative
Comparisons: eclip.shinyapps.io/NormativeComparisons/
- 77. Hammer Ø, Harper DAT, Ryan PD. 2001 PAST: Paleontological Statistics Software Package for
Education and Data Analysis. *Palaeontol. Electron.* **4**, 1-9.
- 78. Anders U, von Koenigswald W, Ruf I, Smith BH. 2011 Generalized individual dental age stages
for fossil and extant placental mammals. *PalZ* **85**, 321-39. (doi: 10.1007/s12542-011-0101-5)
- 79. Seo H, Kim J, Seomun H, Hwang JJ, Jeong HG, Kim JY, Kim HJ, Cho SW. 2017 Eruption of
posterior teeth in the maxilla and mandible for age determination of water deer. *Arch. Oral Biol.* **73**,
237-242. (doi: 10.1016/j.archoralbio.2016.10.020)
- 80. Rüttimeyer, L. 1881 *Beiträge zu einer natürlichen Geschichte der Hirsche*. Zürcher und Furrer. (In
German)
- 81. Groves C. 2016 Systematics of the Artiodactyla of China in the 21st century. *Zool. Res.* **37**, 119-125.
- 82. Wang Y, Harris R. 2015 *Moschus berezovskii*. The IUCN Red List of Threatened Species 2015:
e.T13894A103431781. <http://dx.doi.org/10.2305/IUCN.UK.2015-4.RLTS.T13894A61976926.en>
- 83. Leslie Jr. DM, Lee DN, Dolman RW. 2013 *Elaphodus cephalophus* (Artiodactyla: Cervidae). *Mamm.*
*Species* **45**, 80-91. (doi:10.1644/904.1)
- 84. Hooijer DA. 1958 Fossil Bovidae from the Malay Archipelago and the Punjab. *Zoologische*
*Verhandelingen*, **38**, 1-112.
- 85. Dykoski CA, Edwards RL, Cheng H, Yuan D, Cai Y, Zhang M, Lin Y, Qing J, An Z, Revenaugh J.
2005 A high-resolution, absolute-dated Holocene and deglacial Asian monsoon record from
Dongge Cave, China. *Earth Planet. Sci. Lett.* **233**, 71-86. (doi: 10.1016/j.epsl.2005.01.036)
- 86. Rabett R, Appleby J, Blyth A, Farr L, Gallou A, Giffiths T, Hawkes J, Marcus D, Marlow L, Morley
685 M, Tân NC, Son NV, Penkman K, Reynolds T, Stimpson, C, Szabó K. 2011 Inland shell midden
site-formation: investigation into a late Pleistocene to early Holocene midden from Trảng An,
northern Vietnam. *Quat. Int.* **239**, 153-169. (doi: 10.1016/j.quaint.2010.01.025)
- 87. Guo G, Zhang E. 2005 Diet of the Chinese water deer (*Hydropotes inermis*) in Zhoushan
Archipelago, China. *Acta Theriol. Sin.* **25**, 122-130.
- 88. Kim BJ, Lee NS, Lee SD. 2011 Feeding diets of the Korean water deer (*Hydropotes inermis*
*argyropus*) based on a 202 bp rbcL sequence analysis. *Conserv. Genet.* **12**, 851-856.
(doi:10.1007/s10592-011-0192-2)
- 89. Kim BJ, Oh DH, Chun SH, Lee SD. 2011 Distribution, density, and habitat use of the Korean water
deer (*Hydropotes inermis argyropus*) in Korea. *Landsc. Ecol. Eng.* **7**, 291-297. (doi:10.1007/s11355-010-
0127-y)
- 90. McAdams C, Morley MW, Fu X, Kandyba AV, Derevianko AP, Nguyen DT, Doi NG, Roberts RG.
2020 The Pleistocene geoarchaeology and geochronology of Con Moong Cave, North Vietnam: site
formation processes and hominin activity in the humid tropics. *Geoarchaeology* **35**, 72-97.
- 91. Huong NTM, Hai PV. 2009 Pollen and spore recorded at Con Moong Site (Thanh Hóa). *Vietnam*
*Archaeology* **4**, 24-31.
- 92. Tanabe S, Hori K, Saito Y, Haruyama S, Doanh LQ, Sato Y, Hiraide S. 2003 Sedimentary facies
and radiocarbon dates of the Nam Dinh-1 core from the Song Hong (Red River) delta, Vietnam. *J.*
*Asian Earth Sci.* **21**, 503-513. (doi: 10.1016/S1367-9120(02)00082-2)
- 93. Duong NT, Lieu NTH, Cuc NTT, Saito Y, Huong NTM, Phuong NTM, Thuy AT. 2020 Holocene
paleoshoreline changes of the Red River Delta, Vietnam. *Rev. Palaeobot. Palynol.* **278**, 104235. (doi:
10.1016/j.revpalbo.2020.104235)
- 94. Li Z, Saito Y, Matsumoto E, Wang Y, Haruyama S, Hori K, Doanh LQ. 2006 Palynological record
of climate change during the last deglaciation from the Song Hong (Red River) delta, Vietnam.
*Palaeogeogr. Palaeoclimatol. Palaeoecol.* **235**, 406-430. (doi:10.1016/j.palaeo.2005.11.023)
- 95. Rössner GE, Costeur L, Scheyer TM. 2020 Antiquity and fundamental processes of the antler cycle
in Cervidae (Mammalia). *The Science of Nature* **108**, 3 (doi:10.1101/2020.07.17.208587).

96. Dodson J, Li J, Lu F, Zhang W, Yan H, Cao S. 2019 A Late Pleistocene and Holocene vegetation and environmental record from Shuangchi Maar, Hainan Province, South China. *Palaeogeogr. Palaeoclimatol. Palaeoecol.* **523**, 89-96. (doi: 10.1016/j.palaeo.2019.03.026)

List of Figures

Figure 1. East Asia showing location of Tràng An (red triangle: this study) and distributions of *Hydropotes inermis*: extant range (yellow shading [29]); range in 20th century and estimated maximum westward extent of historical range (orange shading and dashed yellow line, respectively [30]); Holocene records (white circles [11, 32-36]); Upper Pleistocene records (black diamonds [38-43]) and Middle Pleistocene records (black squares [38, 44-48]). Data in table S1. Map (ESRI Satellite base map, EPSG: 3857-WGS 84 Pseudo-Mercator projection) produced in QGIS [49].

Figure 2. Hang Thung Binh 1. (a) Tràng An karst in plan, with location of Hang Thung Binh 1 (TB1 – red triangle) (b) looking north across the coastal plain, towards isolated hill containing TB1 (red triangle) with the Tràng An karst to the east and pagoda complex in the background (original photograph: TK) (c) TB1 in plan showing location of trenches and section D (d) representative west-facing section, showing calibrated radiocarbon dates (cal. BP) and find levels of *Hydropotes inermis* specimens SF40, SF42, SF43 and SF44. Levels are metres above sea level.

Figure 3. Upper Pleistocene specimens of *Hydropotes inermis* recovered from the Hang Thung Binh 1 archaeological cave site in the Tràng An World Heritage Area, Ninh Binh, Northern Vietnam. SF40 right mandible with tooththrow, p2-m3, (a) labial side (b) lingual side and (c) occlusal surface of teeth. SF42 right mandibular fragment with p4 and m1 in situ, (d) labial side (e) lingual side and (f) occlusal surface of teeth. SF43 left maxillary canine (g) lateral and (h) medial aspects. SF44 left maxillary canine tip (i) lateral and (j) medial aspects. Scale = 20 mm.

Figure 4. (a) Tooththrow lengths (p2-m3) and p4 lengths of SF40 and SF42 shown with equivalent data from three taxa, *Hydropotes inermis*, *Moschus moschiferus* and *Elaphodus cephalophus* (tables S3-S5) (b) Details of occlusal surface of teeth of the lower jaw of comparative taxa showing characters referred to in the text. All specimens are at individual dental age stage – IDAS 3 [78]. Specimen details: *Muntiacus muntjac* (OUMNH.ZC-20196); *Moschus moschiferus* (OUMNH.ZC-2894) *Hydropotes inermis* (redrawn from [79]) *Elaphodus cephalophus* (BMNH 92.7.13.1).

List of Tables

Table 1 Provenance and description of identified specimens of *Hydropotes inermis* from the Hang Thung Binh 1 archaeological cave site in the Tràng An World Heritage Landscape Complex Site, Ninh Binh, Northern Vietnam.

Table 2 Measurements of three *Hydropotes inermis* specimens from the archaeological cave site of Hang Thung Binh (TB1) in the Tràng An World Heritage Area, Ninh Binh, Northern Vietnam. All measurements are in mm. C – maxillary canine; p – mandibular premolar; m – mandibular molar; L – length (anterior-posterior); W – width (labial-lingual); p2-p4 – mandibular premolar row length; m1-m3 – mandibular molar row length; p2-m3 – mandibular tooththrow length.

Table 3 Summary of statistical output for tests of normality (Shapiro Wilk W) and univariate normative comparisons (modified *t*-test) of specimens from Tràng An and Lang Trang with comparative data (tables S3-S5). Significant results are shown in bold. Taxon data sets: ¹Holocene; ²Pleistocene and Holocene; ³Upper Pleistocene

Figure 1. East Asia showing location of Trảng An (red triangle: this study) and distributions of *Hydropotes inermis*: extant range (yellow shading [29]); range in 20th century and estimated maximum westward extent of historical range (orange shading and dashed yellow line, respectively [30]); Holocene records (white circles [11, 32-36]); Upper Pleistocene records (black diamonds [38-43]) and Middle Pleistocene records (black squares [38, 44-48]). Data in table S1. Map (ESRI Satellite base map, EPSG: 3857-WGS 84 Pseudo-Mercator projection) produced in QGIS [49].

160x113mm (200 x 200 DPI)

Figure 2. Hang Thung Binh 1. (a) Tràng An karst in plan, with location of Hang Thung Binh 1 (TB1 – red triangle) (b) looking north across the coastal plain, towards isolated hill containing TB1 (red triangle) with the Tràng An karst to the east and pagoda complex in the background (original photograph: TK) (c) TB1 in plan showing location of trenches and section D (d) representative west-facing section, showing calibrated radiocarbon dates (cal. BP) and find levels of *Hydropotes inermis* specimens SF40, SF42, SF43 and SF44. Levels are metres above sea level.

248x240mm (200 x 200 DPI)

Figure 3. Upper Pleistocene specimens of *Hydropotes inermis* recovered from the Hang Thung Binh 1 archaeological cave site in the Tràng An World Heritage Area, Ninh Binh, Northern Vietnam. SF40 right mandible with tooththrow, p2-m3, (a) labial side (b) lingual side and (c) occlusal surface of teeth. SF42 right mandibular fragment with p4 and m1 in situ, (d) labial side (e) lingual side and (f) occlusal surface of teeth. SF43 left maxillary canine (g) lateral and (h) medial aspects. SF44 left maxillary canine tip (i) lateral and (j) medial aspects. Scale = 20 mm.

162x107mm (300 x 300 DPI)

Figure 4. (a) Tooththrow lengths (p2-m3) and p4 lengths of SF40 and SF42 shown with equivalent data from three taxa, *Hydropotes inermis*, *Moschus moschiferus* and *Elaphodus cephalophus* (tables S3-S5) (b) Details of occlusal surface of teeth of the lower jaw of comparative taxa showing characters referred to in the text.

All specimens are at individual dental age stage – IDAS 3 [78]. Specimen details: *Muntiacus muntjak* (OUMNH.ZC-20196); *Moschus moschiferus* (OUMNH.ZC-2894) *Hydropotes inermis* (redrawn from [79]) *Elaphodus cephalophus* (BMNH 92.7.13.1).

139x164mm (200 x 200 DPI)

Table 1 Provenance and description of identified specimens of *Hydropotes inermis* from the Hang Thung Binh 1 archaeological cave site in the Tràng An World Heritage Landscape Complex Site, Ninh Binh, Northern Vietnam.

Site	Trench	Grid sq.	Context	SF no.	Description
TB1	2	TR2EE	F908.2	40	right mandibular body and complete toothrow p2-m3
TB1	2	MS-E	F912	42	right mandible fragment with p4 and m1
TB1	2	MS-E	F912	43	left maxillary canine with root and crown.
TB1	2	MS-E	F912	44	fragment of left maxillary canine - tip of crown

Table 2 Measurements of three *Hydropotes inermis* specimens from the archaeological cave site of Hang Thung Binh (TB1) in the Tràng An World Heritage Area, Ninh Binh, Northern Vietnam. All measurements are in mm. C – maxillary canine; p – mandibular premolar; m – mandibular molar; L – length (anterior-posterior); W – width (labial-lingual); p2-p4 – mandibular premolar row length; m1-m3 – mandibular molar row length; p2-m3 – mandibular tooththrow length.

tooth/toothrow	dimension	SF40	SF42	SF42
C	W	/	/	10.98
p2	L	6.56	/	/
	W	3.19	/	/
p3	L	8.2	/	/
	W	5.36	/	/
p4	L	8.64	9.44	/
	W	5.86	6.24	/
m1	L	9.82	9.8	/
	W	7.2	7.38	/
m2	L	11	/	/
	W	7.11	/	/
m3	L	13.8	/	/
	W	6.68	/	/
p2-p4	L	24.2	/	/
m1-m3	L	34.45	/	/
p2-m3	L	57.96	/	/

Table 3 Summary of statistical output for tests of normality (Shapiro Wilk W) and univariate normative comparisons (modified *t*-test) of specimens from Tràng An and Lang Trang with comparative data (tables S3-S5). Significant results are shown in bold. Taxon data sets: ¹Holocene; ²Pleistocene and Holocene; ³Upper Pleistocene

Spec	Measure.	Taxon	n	W	p (norm.)	hyp	sig	diff	mod. t	p
SF40	L p2-m3	H. inermis ³	28	0.9398	0.1093	2-tailed	N	1.778	1.747	0.084
SF40	L p2-m3	E. cephalophus ¹	17	0.9215	0.1566	1-tailed (smaller)	Y	-1.972	-1.916	0.034
SF40	L p2-m3	M. moschiferus ²	10	0.9287	0.4353	1-tailed (larger)	Y	5.025	4.791	<0.001
SF40	L p2-p4	H. inermis ¹	24	0.9297	0.09588	2-tailed	N	0.78	0.765	0.454
SF40	L p4	E. cephalophus ¹	7	0.9312	0.5611	1-tailed (smaller)	Y	-3.539	-3.31	0.008
SF40	L p4	H. inermis ¹	7	0.9457	0.6902	2-tailed	N	0.261	0.244	0.826
SF42	L p4	H. inermis ¹	7	0.9457	0.6902	2-tailed	N	1.384	1.295	0.23
SF42	L p4	E. cephalophus ¹	7	0.9312	0.5611	1-tailed (smaller)	N	-0.545	-0.51	0.294
SF42	L p4	M. moschiferus ²	7	0.9261	0.518	1-tailed (larger)	Y	6.326	5.917	<0.001
SF43	Cw	H. inermis ¹	8	0.9184	0.4174	2-tailed	N	-0.061	-0.058	0.943
SF40	L p2-m3	H. inermis ¹	28	0.9398	0.1093	1-tailed (larger)	Y	1.778	1.747	0.047
SF40	L p2-p4	H. inermis ¹	24	0.9297	0.09588	1-tailed	N	0.78	0.765	0.224
SF42	L p4	H. inermis ¹	7	0.9457	0.6902	1-tailed	N	1.384	1.295	0.118
SF43	Cw	H. inermis ¹	8	0.9184	0.4174	2-tailed	N	-0.061	-0.058	> 0.9
SF40	L m3	E. cephalophus ³	24	0.9499	0.2696	1-tailed	Y	-4.038	-3.957	<0.001
PIN 5792 /20	L p4	H. inermis ¹	7	0.9457	0.6902	2-tailed	Y	4.276	4	<0.001
PIN 5792 /20	L p4	H. inermis ²	9	0.9318	0.4983	2-tailed	Y	4.177	3.963	0.01

Appendix D

Dear Dr Lindsey,

Re: RSOS-210529 Stimpson et al. Confirmed archaeological evidence of water deer in Vietnam: relics of the Pleistocene or a shifting baseline?

Pleased find enclosed details of the revisions to our manuscript, as requested. We would like to express our thanks to the editors and expert reviewers for their time, constructive criticism and encouraging comments. We feel that the article has benefited and improved as a result of their input and by addressing their comments. We have amended the Acknowledgements: "We thank two anonymous reviewers for their encouraging comments and constructive criticism, which improved the content and structure of the manuscript."

Here, we address the **main points** that you raise, followed by the **point-by-point** responses to each reviewer's comments.

MAIN POINTS

Associate Editor Comments to Author (Dr Emily Lindsey):

Associate Editor: 1

Comments to the Author:

The article was reviewed by two experts, both of whom noted the interest and value of the article and recommended publishing with minor revisions. The two main concerns the reviewers raised that should be addressed for publication were:

1. A more well-developed description of the tooth morphology, and ensuring accuracy and consistency with the diagnostic references.
2. A restructuring of the discussion for clarity and highlighting the importance of the presented study for broader research and conservation-related questions.

1. We have expanded descriptions of lower tooththrows, with greater coverage of p2 and p3 and lower molars. Dental terminology has also been reviewed throughout and modified, where necessary, as directed by the reviewer. **Figure 4** has been revised, accordingly and **Figure 3** has been amended to include occlusal details of the archaeological specimens at greater magnification. Full details are provided in the **point-by-point** responses to **reviewer 2**.

2. We have restructured the discussion to follow the recommendations of the reviewer 1, below. For the sake of transparency, text that has been added or removed as part of the restructure has been highlighted. Full details are provided in the point-by-point responses to **reviewer 1**.

Reviewer 1

"My biggest concern was with the structure of the discussion. It was a little hard to follow, and did not seem, to me at least, to flow well in an "answering research questions" framework. This is not to say that the discussion needs a re-write, but I would say that a re-structuring would benefit the reader. For example, lines 378-380 seem to present a good question that can be addressed with your new data, and so do lines 412-414. I think leading with what you found, i.e., Vietnamese water deer in the latest Pleistocene, and how that can inform the idea of shifting baselines or Quaternary extinctions/extirpations would provide a more impactful discussion. It would also harken back to your introduction where you introduce these ideas at the beginning. Your discussion about the changing floral regimes through the late Pleistocene and Holocene could follow, providing supporting arguments for why the water deer are found in their extant range. While reading the paper, it wasn't really clear to

me how the climatic changes that occurred regionally, i.e. Heinrich event 1, etc, were related to the patterns of vegetation change, and therefore the water deer. It might be worth simplifying that section and making it more streamlined.

Agreed. As a summary, the discussion was restructured as follows:

Significance of Upper Pleistocene Vietnamese water deer – relics, or longer term, previously unrealized population? (and the significance of regional records on range collapse, etc.)

Are the finds a case for reintroduction? Current records in Vietnam - Pleistocene sites in the Northern Vietnam only- genetic and environmental issues for using Pleistocene records as a basis to propose reintroductions. Environmental context of the water deer of Trang An.

Current evidence indicates cooler (but not necessarily drier) conditions (current available date range potentially incorporates stadial and interstadial conditions)– consistent with modern temperate distribution. Therefore, water deer restricted to temperate climes and restricted to Pleistocene in Vietnam? Sub-tropical records as an example of shifting baselines, which challenge this neat but overly simplistic narrative.

Finally, only confirmed, securely dated records from Vietnam would provide unequivocal evidence of a Holocene population.

I also think the manuscript would benefit from a discussion about Quaternary extinctions in the region, and importantly faunal persistence, range shifts, and metapopulation dynamics. What you have seems to be a relict Pleistocene population that got extirpated. There's a lot of exciting research coming out of the region such as the giant muntjac that you described from Northern Vietnam, the work Sam Turvey has done on population collapses in muntjacs, and the new species of gibbon from China. It would help show that there is still much to be learnt from Pleistocene and Holocene records from this part of the world. Moreover, it would be hugely beneficial to see what other species in the region show similar patterns. I know off-hand that all three species of asian rhinos were once more widely distributed through South Asian, Southern China, and Southeast Asia, but are now relegated to isolated pockets. Ending with your conservation and re-introduction message would in my mind be a good way to wrap up this fascinating story."

Agreed, but these issues are discussed and cited in the Introduction as the context for the current research. They are, however, recalled in the discussion with the additional text:

Page 8 Line 338 ("Changed" pdf) *"But what does these new records from Tràng An represent? Are they simply relicts of the Pleistocene or are the new finds an early indication of a previously unrealised, more recent presence in Vietnam and hence, a shifted baseline? This would not be without precedent. Work on later Quaternary palaeontological and zooarchaeological assemblages has highlighted the vulnerability and scale of range collapse of other mammalian herbivores in East and Southeast Asia [e.g. 10,11]."*

Reviewer 1 point-by-point responses

Original	"Changed"	
Page/Line	Page/Line	Comment/Revision
1/31		"It would be good to include your main conclusions here that follow from your analyses of the paleoenvironment."
	1/30	Response

		"We also examine the environmental context of the water deer of Trảng An before considering if the new finds represent relics of an Upper Pleistocene population, or are an early indication of an unrecognised southerly distribution with possible implications for the conservation of the species in the future." Replaced with "Palaeoenvironmental proxies suggest that the Trảng An water deer occupied cooler, but not necessarily drier, conditions than today. We consider if the specimens represent extirpated Pleistocene populations or indicate a previously unrecognised, longer-standing southerly distribution with possible implications for the conservation of the species in the future."
4/106		Comment/Revision
		"Please specify what you mean by exotic"
	3/101	Response
		"both exotic and extinct forms" replaced with "extinct forms and taxa previously unrecorded from Japan"
6/245		Comment/Revision
		This section can be removed or integrated into the discussion
	6/248	Response
		Text removed: Musk deer are currently distributed from the Himalayas to North East Asia, with up to seven extant species [81]. The smallest species, M. berezovskii (Flerov, 1928; extant body weight 6-9 kg), the "dwarf" or "forest musk deer", ranges into South China and marginally into north Vietnam, where it has been reported in karst habitats. The much-reduced Vietnamese populations were, until recently, thought to be Siberian musk deer, M. moschiferus [82]. References updated from [81] onward
		Comment/Revision
6/258		I know what you're trying to say here with statistically significant, but you can remove the term statistically especially since you are referring to Table 3, which has p-values. It reads a lot better when you remove the term statistically
		Response
		"statistically" removed throughout section 4.2 where it is used in this context except from 7/316, where it is stated in a hypothesis
		Comment/Revision
		See R1 main point on restructuring the Dissussion
	9/373	Response
		Text removed (redundant) to accommodate restructuring: "What do we know of the climatic and environmental conditions around TB1 between 13,000- and 16,000-years BP? There is an important environmental question here, as Upper Pleistocene zooarchaeological evidence from cave sites within the Trảng An karst indicate a particular focus on the exploitation of forest-adapted taxa from the karst by humans [56,84]. The position of TB1

		in a “satellite” hill overlooking the plain to the west of the karst “proper”, however, is likely to be pertinent (figure 2).”
7/325		Comment/Revision
		It might be worth giving examples of the climatic instability you are talking about
8/338		Comment/Revision
		Here again, it would be worth talking a bit more about these climatic fluctuations.
		Response
		To address these points, the following text and reference was added as part of the restructured Discussion:
	9/379	“Global climates experienced significant instability between 13,000- and 16,000-years BP [87], which encompassed the glacial conditions of Heinrich Stadial 1 followed by the shift to the relatively brief period of warm and wet conditions of the Bolling-Allerod Interstadial. Chinese speleothem records indicate that this shift occurred in the region towards the end of the more recent radiocarbon date range for the water deer of TB1, between approximately 14, 700 and 13,000 years BP [88]. While it is not possible, at present, to state with certainty if the Tràng An specimens derived from stadial or (at the latter end of the current date range) interstadial conditions, or a transitional period between the two, palaeoenvironmental inferences for the Tràng An karst and the intervening plains can be synthesised from a combination of proxies at various spatial scales.”
	15/709	Additional reference: 88. Zhang H, Ait Brahim Y, Li H, Zhao J, Kathayat G, Tian Y, Baker J, Wang J, Zhang F, Ning Y, Edwards RL. 2019 The Asian summer monsoon: Teleconnections and forcing mechanisms—A review from Chinese speleothem $\delta^{18}O$ records. Quaternary 2, 26. (doi: 10.3390/quat2030026)
9/392		Comment/Revision
		I would cite the authors here instead of calling the article “the 2009 publication”. Or, rephrase the sentence to indicate that in 2009, new dates suggest a post-LGM context
	8/353	Response
		“in the 2009 publication” replaced with “by Popov”

Reviewer 2

“In my eyes the manuscript is almost ready for publication and has major importance to science in the framework described above. The only criticism I come up with concerns the description of the tooth crown morphology. Although the p₄ represents a key tooth for species identification, p₂ and p₃ hold cervid specific differences from Moschus, but aren't described, and m₁ to m₃ are almost not described. This would be important to future work searching for more Hydropotes material in archives and excavations. Please see more detailed comments in the annotated manuscript itself.

We have expanded descriptions of lower tooththrows, with greater coverage of p2, p3 and lower molars. Full details are provided in the point-by-point responses, below.

Moreover, you say that you have used Bärman & Rössner (2011) as the source for tooth crown element nomenclature, what is only true in part. Some elements are differently named than in B & R and do not even follow the principles given there. Hence, either please adapt or clarify in the text that B & R has been used as a basis for your modifications.

Agreed, upon review there are cases where the terminology from older source material (e.g Colbert and Hooijer) has been incorrectly retained. Dental terminology has been reviewed throughout and modified, where necessary, as directed by R2. Full details are provided in the point-by-point responses, below.

The molarisation of p4s is a commonly known phenomenon in mammal odontology and mammal palaeontology what has not to be put in quotation marks. Please see for example Janis and Lister (1985) (attached).

“molarised” is replaced with molarised, throughout

However, although homology of crown elements in non-molarised and molarised is not evidenced in the fossil record by step by step transition states, there are ideas / hypotheses on what element in a non-molarised p4 is homologous to an element in a molarised p4. As to cervids I have provided more details in the annotated manuscript.

Last but not least, as tooth morphology is so crucial in that paper, higher magnified occlusal views are recommended.”

Macro shots of the occlusal surfaces of SF40 and SF42 have been added to figure 3. Figure 4 has also been revised, showing tooththrows at greater magnification and amended with more details of molarised and non-molarised p4s.

Reviewer 2 - point-by-point responses

Original	Changed	
Page/Line	Page/Line	Comment/Revision
1/21		Wouldn't "inappropriately" be more appropriate word instead of "comparatively"?
	1/21	Response
		Agreed. Revised to: "inappropriately"
2/51		Comment/Revision
		begin?
	2/49	Response
		Changed to "begin"
3/127		Comment/Revision
		reflected
	3/120	Response
		Changed to "reflect"
3/127		Comment/Revision
		addition for more specification: "ecological adaptations under different"
	3/120	Response
		"reflected different climatic conditions" revised to "reflect different ecological adaptations under different climatic conditions"

3/128		Comment/Revision
		please add "if"
	3/121	Response
		"if" added
4/149		Comment/Revision
		are composed of?
	4/140	Response
		"comprised" changed to "composed"
		Comment/Revision
5/186		I see differences to Bärmann & Rössner (e.g. in Fig 4 anterior stylid of Bärmann & Rössner became anterior conid and mesolingual conid in B & R became anterior stylid and internal and external postmetacristid in B & R became double fold) and suggest that you either adapt or say that you have modified from Bärmann & Rössner
		Response
		Agreed, upon review there are deviations from Bärmann & Rössner, which retain the nomenclature from earlier, source material. These deviations have been revised and are shown on a point-by-point basis, below. Figure 4 has also been amended and revised, accordingly.
5/205		Comment/Revision
		please add " and pars incisivum" here to complete the list of missing structures
	5/194	Response
		"and pars incisiva" added
		Comment/Revision
5/210		Not true! The dentition is medium worn to slightly worn; even the p2 is slightly worn.
	5/199	Response
		"but is unworn" deleted
		Comment/Revision
5/211		you do not have to put that word in qotation marks as it is a common term in odontology
	5/201	Response
		Quotation marks removed from molarised, here and throughout. Figure 4 is updated.
		Comment/Revision
5/212		Please add "alike the corresponding molars"
	5/202	Response
		"alike the corresponding molars" added
5/213		Comment/Revision
		and elongate in the labial lingual direction
	5/207	Response
		revised to: "the posterior lobe is compressed in the anterior-posterior direction. A well-developed posterior stylid extends lingually from the posterolabial conid.
		Comment/Revision

5/214 and 5/226		homology of structures in non-molarized and molarized premolars is still debated, but from my experience I would say that this is a mesolingual conid, isolated from the transverse crest and shifted a bit anteriorwards, with anterolingual and posterolingual cristid, positioned lingual of the anterior valley. There is no anterior conid. The posterolabial cristid is reduced so that the posterior elements are disconnected from the anterior elements.
		Response
	5/202	Deleted: "The anterior and mesolingual conids are fused, forming a contiguous structure, which enclose an anterior fossette." As per R2's concerns, this statement uses inconsistent terminology and is revised: "The anterior lobe is relatively broad and consists of the mesolingual conid and antero- and posterolingual cristids detached from the transverse crest, shifted to the anterior and positioned to the lingual side of the anterior valley, which recalls an anterior fossa. A deep valley incises the labial side of the tooth at the location of the posterolabial cristid, which demarcates the anterior and posterior lobes. The posterior lobe is compressed in the anterior-posterior direction. A well-developed posterior stylid extends to the lingual side from the posterolabial conid."
		Comment/Revision
5/216		What are "lingual edges of the metaconids" (this is not a term from Bärmann & Rössner)? Postmetacristid and premetacristid?
	5/215	Response
	5/215	Agreed, this recalls the older, source material and does not follow Bärmann & Rössner. Revised to "external postmetacristids" throughout e.g.
	5/226	The molars lack external postmetacristids. "there is no external postmetacristid"
		Comment/Revision
5/216		This is rather little information on differences in the molars especially the grade of prominence of lingual stylids and columns of metaconid and entoconid. Furthermore, how and if cristids connect to one another. Moreover, the the back fossa of the m3 differs in arrangement. And, I am missing details on differences in p2 and p3. Moschus has rather roundish p2 and p3 whereas the cervids have longish triangular ones.
		Response
	5/200	Agreed, coverage is biased towards the p4. The following text has been revise (for nomenclature) with additions: Added: SF4o "The p2 and p3 are relatively elongate and triangular in outline."

	5/210	"The anterior and posterior lobes of the molars are broadly equal in size and shape, with triangular cusps. The adjoining edge of the posterior lobe is offset, labial of the metastylid; the preentocristid abuts the internal postprotocristid of the anterior lobe, next to the posterior margin of the anterior fossa. Except for a pointed mesostylid on the m2, which extends lingually and curves posteriorly, the lingual stylids are not well developed. Rounded mesostylids and entostylids are more prominent on the m1 and m3. The molars lack external postmetacristids. On the labial side, weakly developed anterior cingulids are present on the m1 and m2. A small, slender anterior ectostylid is present on the m3, which has a rounded hypoconulid and an isolated back fossa (figure 4)."
	5/222	SF42 "As in SF40, the p4 is also molarised, with an elongate posterior lobe and well-developed stylid on the posterolabial conid. A weakly-developed anterior cingulid and a weakly-developed and compressed (in the anterior-posterior direction) ectostylid is present on the m1. As for SF40, the joining edge of the posterior lobe is offset, labially. The lingual stylids are weakly developed and there is no external postmetacristid (figure 4)."
	7/253	In the comparison with Moschus "Moschus is indicated against for SF40 by rather rounded, rectangular p2 and p3, compared to the relatively elongate and triangular equivalents in the archaeological specimen. Characteristics of the molars of Moschus also contrast with those of SF40 and SF42. Firstly, the lingual outlines of the molars in SF40 and SF42 are relatively simple and lack external postmetacristids. Well developed external postmetacristids are key diagnostic characters of Moschus [63,80] (figure 4). Secondly, anterior cingulids and mesostylids are relatively well developed and prominent in comparative specimens from the genus Moschus, unlike the archaeological specimens. Thirdly, the location where the posterior lobe of the molars joins their anterior counterpart is much less offset and much more "in line", lingually, with the location of the metastylid. Finally, the margins of the labial conids are rather angular, with shallow indents in the cristids, compared to the archaeological specimens, which are more rounded and lack indentations."
	5/272	In the comparison with E. cephalophus "Unlike the archaeological material, the posterior columns of the molars (entoconid and hypoconid) appear slightly compressed in comparison to the anterior columns (metaconid and protoconid) with relatively well developed ectostylids between them. Furthermore, mesostylids appear well developed in comparison to metastylid and entostylid."
	5/295	In the comparison with Hydropotes

		"As a rule, metastylids were most prominent of the lingual stylids but prominent mesostylids were found to be variably present. In all cases, however, lingual stylids were not as well developed as in Moschus . The lingual edges of the molars are relatively simple and lack external postmetacristids, as in the archaeological specimens, a key character to distinguish Hydropotes from Moschus [63,80]. The location of where the posterior lobe joins the anterior lobe of the molars is offset labially, as in the archaeological specimens.
5/221		Comment/Revision
		the teeth are even medium worn!
	5/220	Response
		", unworn" deleted
5/225		Comment/Revision
		compressed
	5/224	Response
		"elongated" replaced with "compressed"
5/225		Comment/Revision
		anterior-posterior
		Response
	5/224	"Labial-lingual" replaced by "anterior-posterior"
		Comment/Revision
5/230		What about the anterior view? Those elongated ruminant canines are specific in anterior view as well.
	5/230	Response
		Added: "From the anterior aspect, the crown flares slightly to the lateral side and is relatively straight with a slight, sinuous line." Figure 3 has been amended and includes anterior and posterior views of SF ₄₃
5/230		Comment/Revision
		Please add "inlateral view"
	6/232	Response
		"in lateral view" added
5/230		Comment/Revision
		Hmm, strange normally they are convex, may be drop shaped in diameter. Are you sure you haven't mixed up convex with concave?
		Response
	6/232	Thank you! A clumsy error. Revised to: "convex"
5/232		Comment/Revision
		Ahhh, please see my comment two lines above: Iguess there you have mixed up convex with concave.
		Response
		Agreed, as above
6/235		Comment/Revision
		Please add "a"
	6/238	Response
		"a" added
6/235		Comment/Revision
		please add "mesodont"
	6/238	Response

		"mesodont" added
6/236		Comment/Revision
		I don't think this sentence is necessary.
	6/239	Response
		Deleted "The relatively long, posteriorly curved canine with a closed root also indicates the so-called "fanged" deer or musk deer."
6/240		Comment/Revision
		As for 5/214
	6/243	Response
		Deleted: where the anterior and mesolingual conids meet (but are not fused) replaced with "anterolingual cristid of the mesolingual conid adjoins the anterior conid to encircle the anterior valley (but are not fused)"
6/251		Comment/Revision
		As for 5/216
	6/256	Response
		"double-folded" lingual margins of the metaconids"
		Replaced by "well developed external postmetacristids"
		Comment/Revision
6/269		Do you mean that the anterolingual cristid of the mesolingual conid and the anterior stylid limit a short valley ?
	6/277	Response
		"fold in the lingual wall between the anterior stylid and anterior conid, which is absent in"
		Replaced by
		"There is a relatively wide valley in the lingual wall between the anterolingual cristid of the mesolingual conid and an anterior stylid not present in SF40 and SF42 (figure 4)."
7/281		Comment/Revision
		compressed
	7/290	Response
		"elongate" replaced with "compressed"
7/283		Comment/Revision
		Comparison with the anterior lobe of the p4 is missing.
	9/292	Response
		Added "The metaconid of the anterior lobe is isolated from the transverse crest and shifted anteriorly to the lingual side of the anterior valley. The labial side of the tooth is deeply incised at the location of the posterolabial cristid."
8/327		Comment/Revision
		by
		Response
		This sentence has been removed as a result of restructuring the Discussion (see R1 – main points)
		Comment/Revision
8/348		Are those deep rooting trees? I guess in karst plants have to root deep to get enough water.

	9/396	Response
		All those taxa are wind pollinated, so it is possible those grains could have come from forests on valley floors rather than, or in addition to, the karst limestone. To reflect this "and intervening valleys" is included: "These data suggest that limestone karsts and intervening valleys on the southern margin of the Song Hong delta remained forested throughout the last deglaciation."
9/392		Comment/Revision
		Last Glacial Maximum
	8/353	Response
		"LGM" replaced with "Last Glacial Maximum"
Figure 2 (a)		Comment/Revision
		numbers are hard to see, should be improved
		Response
		Numbers made more visible
Figure 2 (d)		Comment/Revision
		Not indicated in legend
		Response
		The limestone is indicated, but the basal rock in the figure lacks grey shading, as per the legend and higher in the sequence – this is corrected.
		Comment/Revision
Figure 3		As p₄ morphology plays a crucial role in identification, additional figures of the occlusal surface of both p₄s with a higher magnification were eligible.
		Response
		Higher magnification images of the details of the occlusal surface of the p ₄ have been added to figures 3 and 4.
Figure 3 caption		Comment/Revision
		(e)
		Response
		"I" replaced by "(e)": note, caption revised with updated figure.
		Comment/Revision
		Why don't you give canin measurements? This would be important for future studies.
		Response
		Agreed, but our specimens were not complete. We provide measurements in the description and include available comparative data in the supplementary files.
		Comment/Revision
Table 2		Why don't you give length as well? Both measurements would be more indicative.
		Response
		We now include a length measurement for SF ₄₃ , with the caveat it is broken. The table caption is updated accordingly.

Appendix E

Dear Dr Lindsey,

Re: RSOS-210529 Stimpson et al. Confirmed archaeological evidence of water deer in Vietnam: relics of the Pleistocene or a shifting baseline?

We would like to express our thanks to the editors and expert reviewers for their time, constructive criticism and encouraging comments. We feel that the article has benefited and improved as a result of their input and by addressing their comments. (This is reflected in an addition to the Acknowledgements: "We thank two anonymous reviewers for their encouraging comments and constructive criticism, which improved the content and structure of the manuscript.")

Here, we address the **main points** that you raise, followed by the **point-by-point** responses to each reviewer's comments.

MAIN POINTS

Associate Editor Comments to Author (Dr Emily Lindsey):

Associate Editor: 1

Comments to the Author:

The article was reviewed by two experts, both of whom noted the interest and value of the article and recommended publishing with minor revisions. The two main concerns the reviewers raised that should be addressed for publication were:

1. A more well-developed description of the tooth morphology, and ensuring accuracy and consistency with the diagnostic references.
2. A restructuring of the discussion for clarity and highlighting the importance of the presented study for broader research and conservation-related questions.

1. We have expanded descriptions of lower tooththrows, with greater coverage of p2 and p3 and lower molars. Dental terminology has also been reviewed throughout and modified, where necessary, as directed by the reviewer. **Figure 4** has been revised, accordingly and **Figure 3** has been amended to include occlusal details of the archaeological specimens at greater magnification. Full details are provided in the **point-by-point** responses to **reviewer 2**.

2. We have restructured the discussion to follow the recommendations of the reviewer 1, below. For the sake of transparency, text that has been added or removed as part of the restructure has been highlighted. Full details are provided in the point-by-point responses to **reviewer 1**.

Reviewer 1

"My biggest concern was with the structure of the discussion. It was a little hard to follow, and did not seem, to me at least, to flow well in an "answering research questions" framework. This is not to say that the discussion needs a re-write, but I would say that a re-structuring would benefit the reader. For example, lines 378-380 seem to present a good question that can be addressed with your new data, and so do lines 412-414. I think leading with what you found, i.e., Vietnamese water deer in the latest Pleistocene, and how that can inform the idea of shifting baselines or Quaternary extinctions/extirpations would provide a more impactful discussion. It would also harken back to your introduction where you introduce these ideas at the beginning. Your discussion about the changing floral regimes through the late Pleistocene and Holocene could follow, providing supporting arguments for why the water deer are found in their extant range. While reading the paper, it wasn't really clear to me how the climatic changes that occurred regionally, i.e. Heinrich event 1, etc, were related to the

patterns of vegetation change, and therefore the water deer. It might be worth simplifying that section and making it more streamlined.

Agreed. As a summary, the discussion was restructured as follows:

Significance of Upper Pleistocene Vietnamese water deer – relics, or longer term, previously unrealized population? (and the significance of regional records on range collapse, etc.)

Are the finds a case for reintroduction? Current records in Vietnam - Pleistocene sites in the Northern Vietnam only- genetic and environmental issues for using Pleistocene records as a basis to propose reintroductions. Environmental context of the water deer of Trang An.

Current evidence indicates cooler (but not necessarily drier) conditions (current available date range potentially incorporates stadial and interstadial conditions)– consistent with modern temperate distribution. Therefore, water deer restricted to temperate climates and restricted to Pleistocene in Vietnam? Sub-tropical records as an example of shifting baselines, which challenge this neat but overly simplistic narrative.

Finally, only confirmed, securely dated records from Vietnam would provide unequivocal evidence of a Holocene population.

I also think the manuscript would benefit from a discussion about Quaternary extinctions in the region, and importantly faunal persistence, range shifts, and metapopulation dynamics. What you have seems to be a relict Pleistocene population that got extirpated. There's a lot of exciting research coming out of the region such as the giant muntjac that you described from Northern Vietnam, the work Sam Turvey has done on population collapses in muntjacs, and the new species of gibbon from China. It would help show that there is still much to be learnt from Pleistocene and Holocene records from this part of the world. Moreover, it would be hugely beneficial to see what other species in the region show similar patterns. I know off-hand that all three species of asian rhinos were once more widely distributed through South Asian, Southern China, and Southeast Asia, but are now relegated to isolated pockets. Ending with your conservation and re-introduction message would in my mind be a good way to wrap up this fascinating story."

Agreed, but these issues are discussed and cited in the Introduction as the context for the current research. They are, however, recalled in the discussion with the additional text:

"But what does these new records from Trảng An represent? Are they simply relicts of the Pleistocene or are the new finds an early indication of a previously unrealised, more recent presence in Vietnam and hence, a shifted baseline? This would not be without precedent. Work on later Quaternary palaeontological and zooarchaeological assemblages has highlighted the vulnerability and scale of range collapse of other mammalian herbivores in East and Southeast Asia [e.g. 10,11]."

Reviewer 1 point-by-point responses

Original	"Changed"	
Page/Line	Page/Line	Comment/Revision
1/31		"It would be good to include your main conclusions here that follow from your analyses of the paleoenvironment."
		Response
		"We also examine the environmental context of the water deer of Trảng An before considering if the new finds represent relicts of an Upper Pleistocene population, or are an early indication of an

		unrecognised southerly distribution with possible implications for the conservation of the species in the future.” Replaced with “Palaeoenvironmental proxies suggest that the Trảng An water deer occupied cooler, but not necessarily drier, conditions than today. We consider if the specimens represent extirpated Pleistocene populations or indicate a previously unrecognised, longer-standing southerly distribution with possible implications for the conservation of the species in the future.”
		Comment/Revision
4/106		“Please specify what you mean by exotic”
		Response
		“both exotic and extinct forms” replaced with “extinct forms and taxa previously unrecorded from Japan”
6/245		Comment/Revision
		This section can be removed or integrated into the discussion
		Response
		Text removed: Musk deer are currently distributed from the Himalayas to North East Asia, with up to seven extant species [81]. The smallest species, M. berezovskii (Flerov, 1928; extant body weight 6-9 kg), the “dwarf” or “forest musk deer”, ranges into South China and marginally into north Vietnam, where it has been reported in karst habitats. The much-reduced Vietnamese populations were, until recently, thought to be Siberian musk deer, M. moschiferus [82]. References updated from [81] onward
		Comment/Revision
6/258		I know what you’re trying to say here with statistically significant, but you can remove the term statistically especially since you are referring to Table 3, which has p-values. It reads a lot better when you remove the term statistically
		Response
		“statistically” removed throughout section 4.2 where it is used in this context except from line 290, where it is stated in a hypothesis
		Comment/Revision
7/325		It might be worth giving examples of the climatic instability you are talking about
		Comment/Revision
8/338		Here again, it would be worth talking a bit more about these climatic fluctuations.
		Response
		The following text and reference was added as part of the restructured Discussion: “Global climates experience significant instability between 13,000- and 16,000-years BP [87], which encompassed the glacial conditions of Heinrich Stadial 1 followed by the shift to the relatively brief

		period of warm and wet conditions of the Bolling-Allerod Interstadial. Chinese speleothem records indicate a regional shift between approximately 14, 700 and 13,000 years BP [88].” This potential significance of this shift is also highlighted: “Given the range that is encompassed by the available radiocarbon dates, it is not possible to state if the specimens of Trảng An specimens derived from stadial or interstadial conditions for certain, but the available proxies data suggest that similar or preferred habitats of extant water deer were present in the Upper Pleistocene on the plains around TB1, which reflected cooler, but not necessarily drier, conditions than the sub-tropical climate of the Holocene.”
		Comment/Revision
9/392		I would cite the authors here instead of calling the article “the 2009 publication”. Or, rephrase the sentence to indicate that in 2009, new dates suggest a post-LGM context
		Response
		“in the 2009 publication” replaced with “by Popov”

Reviewer 2

“In my eyes the manuscript is almost ready for publication and has major importance to science in the framework described above. The only criticism I come up with concerns the description of the tooth crown morphology. Although the p4 represents a key tooth for species identification, p2 and p3 hold cervid specific differences from Moschus, but aren't described, and m1 to m3 are almost not described. This would be important to future work searching for more Hydropotes material in archives and excavations. Please see more detailed comments in the annotated manuscript itself.

We have expanded descriptions of lower tooththrows, with greater coverage of p2, p3 and lower molars. Full details are provided in the point-by-point responses, below.

Moreover, you say that you have used Bärman & Rössner (2011) as the source for tooth crown element nomenclature, what is only true in part. Some elements are differently named than in B & R and do not even follow the principles given there. Hence, either please adapt or clarify in the text that B & R has been used as a basis for your modifications.

Agreed, upon review there are cases where the terminology from older source material (e.g Colbert and Hooijer) has been incorrectly retained. Dental terminology has been reviewed throughout and modified, where necessary, as directed by R2. Full details are provided in the point-by-point responses, below.

The molarisation of p4s is a commonly known phenomenon in mammal odontology and mammal palaeontology what has not to be put in quotation marks. Please see for example Janis and Lister (1985) (attached).

“molarised” is replaced with molarised, throughout

However, although homology of crown elements in non-molarised and molarised is not evidenced in the fossil record by step by step transition states, there are ideas / hypotheses on what element in a non-molarised p4 is homologous to an element in a molarised p4. As to cervids I have provided more details in the annotated manuscript.

Last but not least, as tooth morphology is so crucial in that paper, higher magnified occlusal views are recommended."

Macro shots of the occlusal surfaces of SF40 and SF42 have been added to figure 3. Figure 4 has also been revised, showing toothrows at greater magnification and amended with more details of molarised and non-molarised p4s.

Reviewer 2 - point-by-point responses

Original Page/Line	Revised Page/Line	Comment/Revision
1/21		Wouldn't "inappropriately" the more appropriate word instead of "comparatively"?
		Response
		Agreed. Revised to: "inappropriately"
2/51		Comment/Revision
		begin?
		Response
		Changed to "begin"
3/127		Comment/Revision
		reflected
		Response
		Changed to "reflect"
		Comment/Revision
3/127		addition for more specification: "ecological adaptations under different"
		Response
		"reflected different climatic conditions" revised to "reflect different ecological adaptations under different climatic conditions"
3/128		Comment/Revision
		please add "if"
		Response
		"if" added
4/149		Comment/Revision
		are composed of?
		Response
		"comprised" changed to "composed"
		Comment/Revision
5/186		I see differences to Bärmann & Rössner (e.g. in Fig 4 anterior stylid of Bärmann & Rössner became anterior conid and mesolingual conid in B & R became anterior stylid and internal and external postmetacristid in B & R became double fold) and suggest that you either adapt or say that you have modified from Bärmann & Rössner
		Response
		Agreed, upon review there are deviations from Bärmann & Rössner, which retain the nomenclature from earlier, source material. These deviations have been revised and are shown on a point-by-point basis, below (from Page 5, line 214 onwards). Figure 4 has also been amended and revised, accordingly.
5/205		Comment/Revision

		please add " and pars incisivum" here to complete the list of missing structures
		Response
		"and pars incisiva" added
		Comment/Revision
5/210		Not true! The dentition is medium worn to slightly worn; even the p2 is slightly worn.
		Response
		"but is unworn" deleted
		Comment/Revision
5/211		you do not have to put that word in qotation marks as it is a common term in odontology
		Response
		Quotation marks removed from molarised, here and throughout. Figure 4 is updated.
		Comment/Revision
5/212		Please add "alike the corresponding molars"
		Response
		"alike the corresponding molars" added
5/213		Comment/Revision
		and elongate in the labial-lingual direction
		Response
		revised to: "the posterior lobe is compressed in the anterior-posterior direction. A well-developed posterior stylid extends lingually from the posterolabial conid.
		Comment/Revision
5/214 and 5/226		homology of structures in non-molarized and molarized premolars is still debated, but from my experience I would say that this is a mesolingual conid, isolated from the transverse crest and shifted a bit anteriorwards, with anterolingual and posterolingual cristid, positioned lingual of the anterior valley. There is no anterior conid. The posterolabial cristid is reduced so that the posterior elements are disconnected from the anterior elements.
		Response
		Deleted: "The anterior and mesolingual conids are fused, forming a contiguous structure, which enclose an anterior fossette." As per R2's concerns, this statement uses inconsistent terminology and is revised: "The anterior lobe is relatively broad and consists of the mesolingual conid and antero- and posterolingual cristids detached from the transverse crest, shifted to the anterior and positioned to the lingual side of the anterior valley, which recalls an anterior fossa. A deep valley incises the labial side of the tooth at the location of the posterolabial cristid, which demarcates the anterior and posterior lobes. The posterior lobe is compressed in the anterior-posterior direction. A well-developed posterior stylid extends to the lingual side from the posterolabial conid."

		Comment/Revision
5/216		What are "lingual edges of the metaconids" (this is not a term from Bärmann & Rössner)? Postmetacristid and premetacristid?
		Response
		Agreed, this recalls the older, source material and does not follow Bärmann & Rössner. Revised to: The molars lack external postmetacristids.
		Comment/Revision
5/216		This is rather little information on differences in the molars especially the grade of prominence of lingual stylids and columns of metaconid and entoconid. Furthermore, how and if cristids connect to one another. Moreover, the the back fossa of the m3 differs in arrangement. And, I am missing details on differences in p2 and p3. Moschus has rather roundish p2 and p3 whereas the cervids have longish triangular ones.
		Response Agreed, coverage is biased towards the p4. The following text has been added: Added: SF40 "The p2 and p3 are relatively elongate and triangular in outline." "The anterior and posterior lobes of the molars are broadly equal in size and shape, with triangular cusps. The adjoining edge of the posterior lobe is offset, labial of the metastylid; the preentocristid abuts the internal postprotocristid of the anterior lobe, next to the posterior margin of the anterior fossa. The lingual edges of the molars are relatively simple. Except for a pointed mesostylid on the m2, which extends lingually and curves posteriorly, the lingual stylids are not well developed. Rounded mesostylids and entostylids are more prominent on the m1 and m3. The molars lack external postmetacristids. On the labial side, weakly developed anterior cingulids are present on the m1 and m2. A small, slender anterior ectostylid is present on the m3, which has a rounded hypoconulid and entoconulid and an isolated back fossa (figure 4)." SF42 "As in SF40, the p4 is also molarised, with an elongate posterior lobe and well-developed stylid on the posterolabial conid. A weakly-developed anterior cingulid and a weakly-developed and compressed (in the anterior-posterior direction) ectostylid is present on the m1. As for SF40, the joining edge of the posterior lobe is offset, labially. The lingual stylids are weakly developed and there is no external postmetacristid (figure 4)." In the comparison with Moschus "Moschus is indicated against for SF40 by rather rounded, rectangular p2 and p3, compared to the relatively elongate and

		triangular equivalents in the archaeological specimen. Characteristics of the molars of Moschus also contrast with those of SF40 and SF42. Firstly, the lingual outlines of the molars in SF40 and SF42 are relatively simple and lack external postmetacristids. Well developed external postmetacristids are key diagnostic characters of Moschus [63,80] (figure 4). Secondly, anterior cingulids and mesostylids are relatively well developed and prominent in comparative specimens from the genus Moschus, unlike the archaeological specimens. Thirdly, the location where the posterior lobe of the molars joins their anterior counterpart is much less offset and much more “in line”, lingually, with the location of the metastylid. Finally, the labial margins of the labial conids are rather angular, with shallow indents in the cristids, compared to the archaeological specimens, which are more rounded and lack indentations.” In the comparison with E. cephalophus “Unlike the archaeological material, the posterior columns of the molars (entoconid and hypoconid) appear slightly compressed in comparison to the anterior columns (metaconid and protoconid) with relatively well developed ectostylids between them. Furthermore, mesostylids appear well developed in comparison to metastylid and entostylid.” In the comparison with Moschus “more rectangular, rounded p2 and p3 in musk deer and” In the comparison with Hydropotes “As a rule, metastylids were most prominent of the lingual stylids but prominent mesostylids were found to be variably present. In all cases, however, lingual stylids were not as well developed or as robust as in Moschus. The lingual edges of the molars are relatively simple and lack external postmetacristids, as in the archaeological specimens, a key character to distinguish Hydropotes from Moschus [63,80]. The location of where the posterior lobe joins the anterior lobe of the molars is offset labially, as in the archaeological specimens.
5/221		Comment/Revision
		the teeth are even medium worn!
		Response
		“, unworn” deleted
5/225		Comment/Revision
		compressed
		Response
		“elongated” replaced with “compressed”
5/225		Comment/Revision
		anterior-posterior
		Response
		“Labial-lingual” replaced by “anterior-posterior”
		Comment/Revision

5/230		What about the anterior view? Those elongated ruminant canines are specific in anterior view as well.
		Response
		Added: "The anterior aspect of the crown flares slightly to the lateral side and is relatively straight, with a slight, sinuous line. " Figure 3 has been amended and includes anterior and posterior views of SF43
		Comment/Revision
5/230		Please add "inlateral view"
		Response
		"in lateral view" added
5/230		Comment/Revision
		Hmm, strange normally they are convex, may be drop shaped in diameter. Are you sure you haven't mixed up convex with concave?
		Response
		Thank you! A clumsy error. Revised to: "convex"
5/232		Comment/Revision
		Ahhh, please see my comment two lines above: I guess there you have mixed up convex with concave.
		Response
		Agreed, as above
6/235		Comment/Revision
		Please add "a"
		Response
		"a" added
6/235		Comment/Revision
		please add "mesodont"
		Response
		"mesodont" added
6/236		Comment/Revision
		I don't think this sentence is necessary.
		Response
		Deleted "The relatively long, posteriorly curved canine with a closed root also indicates the so-called "fanged" deer or musk deer."
6/240		Comment/Revision
		As for 5/214
		Response
		Deleted: where the anterior and mesolingual conids meet (but are not fused) replaced with "anterolingual cristid of the mesolingual conid adjoins the anterior conid to encircle the anterior valley (but are not fused)"
6/251		Comment/Revision
		As for 5/216
		Response
		"double-folded" lingual margins of the metaconids" Replaced by "well developed external postmetacristids"
		Comment/Revision

6/269		Do you mean that the anterolingual cristid of the mesolingual conid and the anterior stylid limit a short valley ?
		Response
		"fold in the lingual wall between the anterior stylid and anterior conid, which is absent in" Replaced by "there is a relatively wide valley in the lingual wall between the anterolingual cristid of the mesolingual conid and an anterior stylid not present in SF40 and SF42 (figure 4)."
7/281		Comment/Revision
		compressed
		Response
		"elongate" replaced with "compressed"
7/283		Comment/Revision
		Comparison with the anterior lobe of the p4 is missing.
		Response
		Added "The metaconid of the anterior lobe is isolated from the transverse crest and shifted anteriorly to the lingual side of the anterior valley. The labial side of the tooth is deeply incised at the location of the posterolabial cristid."
8/327		Comment/Revision
		by
		Response
		This sentence has been removed as a result of the restructured of Discussion.
		Comment/Revision
8/348		Are those deep rooting trees? I guess in karst plants have to root deep to get enough water.
		Response
		All those taxa are wind pollinated, so it is possible those grains could have come from forests on valley floors rather than, or in addition to, the karst limestone. To reflect this "and intervening valleys" is included: "These data suggest that limestone karsts and intervening valleys on the southern margin of the Song Hong delta remained forested throughout the last deglaciation."
9/392		Comment/Revision
		Last Glacial Maximum
		Response
		"LGM" replaced with "Last Glacial Maximum"
Figure 2 (a)		Comment/Revision
		numbers are hard to see, should be improved
		Response
		Numbers made more visible
Figure 2 (d)		Comment/Revision
		Not indicated in legend
		Response

		The limestone is indicated, but the basal rock in the figure lacks grey shading, as per the legend and higher in the sequence – this is corrected.
		Comment/Revision
Figure 3		As p4 morphology plays a crucial role in identification, additional figures of the occlusal surface of both p4s with a higher magnification were eligible.
		Response
		Higher magnification images of the details of the occlusal surface of teeth have been added to figures 3 and 4.
Figure 3 caption		Comment/Revision
		(e)
		Response
		"l" replaced by "(e)": note, caption revised with updated figure.
		Comment/Revision
		Comment/Revision
		Why don't you give canin measurements? This would be important for future studies.
		Response
		Agreed, but our specimens were not complete. We provide measurements in the description and include available comparative data in the supplementary files.
		Comment/Revision
Table 2		Why don't you give length as well? Both measurements would be more indicative.
		Response
		We now include a length measurement for SF43, with the caveat it is broken. The table caption is updated accordingly.